# SURVIVAL VAE: ROBUST LOCAL EXPLANATIONS VIA DOUBLE-PASS RISK CONSISTENCY

## ABSTRACT

In the era of advanced machine learning, the need to explain models has grown significantly. One particular domain, survival analysis, has benefited from the rise of deep learning but has lagged behind in the development of methods for explaining risk and survival models. Only a few works have adapted explainable AI methods, such as LIME and SHAP, to survival analysis. Despite these efforts, explaining survival models remains challenging given the complex nature of the data used for survival predictions and the presence of censoring. In this work, we propose a local feature identification method that inherently operates on the instance ordering induced by event and censoring times. It enables faithful, per-sample feature importance by identifying which reconstructed input features preserve consistency in predicted survival risk across a double-pass through the variational autoencoder. Empirical results on the large high-dimensional multi-cohort dataset from The Cancer Genome Atlas demonstrate superior quantitative performance of our method. Qualitatively, analysis of mask weights highlights the biological relevance of the feature selection process. This information can be used to identify new diagnostic markers and treatment targets for cancer patients.

## 1 INTRODUCTION

The first thing cancer patients want to know once they receive the diagnosis is their prognosis, which is referred to as survival in the context of clinical data analysis. This important metric is used to decide treatment for individual patients, but can also serve as a tool of prioritization when assigning funds to research on new treatments (Royle et al., 2023). In recent decades, it has become clear that cancer is an extremely heterogeneous disease that requires personalized approaches not only for diagnosis and, therefore, survival prognosis, but also for treatment (Hu et al., 2017; Tsimberidou et al., 2020). Sequencing of the patient's DNA or RNA helps to understand the underlying characteristics of each individual tumor (Passaro et al., 2024; Fayyaz et al., 2024). Gene expression data, which are high-dimensional, can be compared between patients and is therefore a rich source of information to predict survival (Giampietri et al., 2022). Survival analysis plays a central role in modeling survival behavior by linking covariates (features) to time-to-event outcomes.

However, censoring poses a challenge for survival modeling; it occurs when a patient drops out or the clinical study ends before the event occurs, leaving the survival time only partially observed. Discarding all censored instances prior to learning leads to information loss, especially when censored samples are dominant. Therefore, survival analysis methods incorporate censored data in the estimation of survival and hazard functions. A widely accepted formulation is the Cox Proportional Hazards (CPH) model (Cox, 1972), which assumes proportional hazards and avoids fully parametric estimation of the hazard curves.

With the increasing success of deep learning, DeepSurv (Katzman et al., 2018) demonstrates how to design a training objective for deep neural networks that respects the Proportional Hazards (PH) assumption. At the same time, the field of eXplainable AI (XAI) has emerged to interpret the predictions of complex models. At the intersection of survival analysis and XAI, notable approaches have been proposed, such as SurvLIME (Kovalev et al., 2020) and SurvSHAP (Krzyziński et al., 2023), to imitate LIME (Ribeiro et al., 2016) and SHAP (Lundberg & Lee, 2017) on model estimates of survival curves. Despite recent advances, most existing approaches apply general XAI methods to the output of survival models, rather than tailoring the explanation process to the characteristics of survival data.

This work presents an attempt to fulfill a major unmet need: designing XAI methods that inherently account for both complete (event) and incomplete (censored) observations when explaining individual risk predictions through a local feature identification method. We devise Double-Pass Survival VAE (DP-SurVAE), which applies masking to reconstructed samples and introduces a novel double-pass objective to ensure risk consistency. We demonstrate theoretically that risk consistency is guaranteed through the double-pass objective. We apply our method to gene expression from RNA (mRNA) and microRNA (miRNA) across a total of 37 disease cohorts from The Cancer Genome Atlas (TCGA) dataset to predict survival for individual patients depending on the cancer type. We show empirically that DP-SurVAE achieves superior quantitative performance compared to other methods. Furthermore, the XAI component allows us to identify features relevant for the predictions by providing faithful feature importance. For gene expression, the results confirm that our method selects key genes relevant to cancer risk prediction, supported by the extensive annotation available for mRNA.

## 2   RELATED WORK: INTERPRETABLE SURVIVAL ANALYSIS

Local Interpretable Model-agnostic Explanations (LIME) is a prominent XAI method that aims to explain a black-box model's prediction by fitting an interpretable local model around the instance to be explained. Lundberg & Lee apply the Shapley value concept from game theory (Kuhn & Tucker, 1953) to estimate the contribution of each feature to an individual model prediction. In their framework, SHAP unifies several additive feature attribution methods, computing Shapley values as marginal contributions based on conditional expectations of the model output. Both LIME and SHAP have been applied and adapted to explain model predictions in survival analysis settings. For example, LIME is utilized by Kasim et al. (2024) to interpret predictions of in-hospital mortality among STEMI patients. Additionally, Moncada-Torres et al. (2021) argue that the disadvantage of advanced models being black boxes—despite outperforming CPH models by capturing nonlinearities and complex interactions—can be mitigated by equipping these models with explanations from SHAP, thereby enhancing their interpretability. SurvLIME adapts LIME on survival data by approximating the survival function using the CPH model locally at the sample to be explained. Similar to LIME, CPH is considered a locally interpretable model due to its log-linear nature. SurvLIME's objective minimizes the distance between the survival function, fitted using the cumulative CPH, and the model's prediction. SurvSHAP extends SHAP to create a time-dependent attribution explanation for each feature; it tries to explain the black-box survival model by computing the Shapley values based on the change in expected survival when conditioning on the studied features. Unlike proportional hazards, the survival function depends on time, which emphasizes the importance of time-dependent explanations. CoxNAM (Xu & Guo, 2023) maximizes the partial likelihood of the Cox proportional hazards (CPH) model, where the uni-log-risk functions are parameterized using Neural Additive Models (NAMs) (Agarwal et al., 2021). In CoxNAM, a dedicated subnetwork is learned for each feature, estimating a one-dimensional shape function. See Appendix A for further details on the use of machine learning for survival analysis.

Despite these advances, existing approaches mainly apply general XAI methods to survival models, rather than tailoring the explanation process to the special characteristics of survival models, that is, that data might be censored. This work presents an attempt to fulfill this major unmet need: designing a local feature identification method that inherently accounts for both complete (event) and incomplete (censored) observations when explaining individual risk predictions.

## 3   BACKGROUND: SURVIVAL ANALYSIS

Survival analysis is the field that models the relationship between instance-level features and the time till the occurrence of a specific event. We use the term instance to refer to subjects such as humans, animals, or mechanical systems. Each instance is represented as $(\boldsymbol{x}_i, t_i, \delta_i)$, where $\boldsymbol{x}_i \in \mathbb{R}^d$ is the feature vector, $t_i$ is the time of event or censoring, and $\delta_i \in \{0, 1\}$ indicates whether the event was observed (1) or not (0) as in censoring. We focus in this work on right-censoring.

The time-to-event random variable $T$ is typically described using three functions: (i) the probability density function (PDF) $f(t)$, (ii) the survival function $S(t)$, and (iii) the hazard function $\lambda(t)$. These are related mathematically, and each one determines the others.

The PDF models the likelihood of an event occurring in an infinitesimal interval: $f(t) = \lim_{\Delta t \to 0} \frac{P(t < T \leq t + \Delta t)}{\Delta t}$. The survival function gives the probability of surviving beyond time $t$: $S(t) = P(T > t) = 1 - F(t) = \int_t^\infty f(x)\,dx$, where $F(t)$ is the cumulative distribution function. The hazard function models the event rate at time $t$ when surviving till $t$: $\lambda(t) = \lim_{\Delta t \to 0} \frac{P(t < T \leq t + \Delta t \mid T > t)}{\Delta t} = \frac{f(t)}{S(t)}$. We use the notation $f(t; \boldsymbol{x})$, $S(t; \boldsymbol{x})$, and $\lambda(t; \boldsymbol{x})$, where $\boldsymbol{x}$ is the covariate vector (feature vector).

The CPH model assumes hazard proportionality between features, and a log-linear risk function:

$$\lambda(t; \boldsymbol{x}) = \lambda_0(t) \cdot \lambda(\boldsymbol{x}) = \lambda_0(t) \cdot \exp\left( \sum_{i=1}^d \beta_i x_i \right), \tag{1}$$

where $\lambda_0(t)$ is the baseline hazard and $\beta_i$ are learned coefficients (Lee, 1992). The Cox model is trained by maximizing the partial likelihood (PL), which considers the ordering of events and includes only uncensored instances. Each event's contribution is conditioned on its risk set—the group of instances still under observation at the event time. The conditional probability for an event occurring at time $t_o$ for instance $(\boldsymbol{x}_o, t_o, \delta_o)$ is given by

$$L_o(\boldsymbol{\beta}) = \frac{\exp\left( \sum_{i=1}^n \beta_i x_{oi} \right)}{\sum_{(\boldsymbol{x}_l, t_l, \delta_l) \in R(t_o)} \exp\left( \sum_{i=1}^n \beta_i x_{li} \right)} \ , \tag{2}$$

where $R(t_o) = \{ (\boldsymbol{x}_l, t_l, \delta_l) \in D \mid t_l \geq t_o \}$ is the risk set at time $t_o$. For a dataset with $m$ uncensored events at distinct times, the partial likelihood is the product over all such instances: $PL(\boldsymbol{\beta}) = \prod_{o=1}^m L_o(\boldsymbol{\beta})$.

### 3.1 Notation for Local Explanations

Following the notation proposed in LIME (Ribeiro et al., 2016), let $f : \mathbb{R}^d \to \mathbb{R}$ be the trained model to be explained. The explanation model $g \in G$ often acts in the domain of simplified samples $x' \in \{0, 1\}^{d'}$. Local explanation methods often try to enforce $g(x') \approx f(h_x(x'))$, where $h_x$ is the mapping function that is specific to the sample $x$ (Lundberg & Lee, 2017). LIME tries to explain the prediction $f$ for an instance $x$ by fitting a human-comprehensible function (e. g., linear model) locally in the neighborhood of $x$. To this end, LIME searches for $g$ that minimizes $\mathcal{L}(f, g, \pi_x) + \Omega(g)$, where $\pi_x$ is a distance measure, $\mathcal{L}$ measures how faithful $g$ approximates $f$ near $x$, and $\Omega(g)$ is the complexity of $g$.

## 4 Methodology

The core of our method is to robustly encode and decode patient vectors using a Variational Auto Encoder (VAE), followed by a masking operation applied to the reconstructed vectors via an attention mechanism. Robustness is enforced by ensuring that the hazard estimated from sparsified (masked) samples satisfies the same negative log-likelihood objective as the original, unmasked data. This design enables a local feature identification method that provides faithful, per-sample feature importance by revealing which input features most strongly influence survival risk predictions. Simply stated and in LIME's terminology, we aim for $f(g(x)) \approx f(x)$, that is, the explained samples (masked and sparsified) should lead to the same survival ranking as the original samples; whereas LIME aims for $g(x') \approx f(h_x(x'))$, that is, the explainer $g$ should approximate the original model on the reduced samples (see Section 3.1). We refer to our method as Double-Pass Survival VAE (DP-SurVAE). Its architecture and workflow are illustrated in Figure 1. The first pass (solid red arrows) encodes the input $\boldsymbol{x}$ into a latent representation $\mu_{\boldsymbol{x}}$, used for both reconstruction and risk prediction. The second pass (dashed red arrows) propagates the masked reconstructed input $\hat{\boldsymbol{x}}$ through the encoder to compute a latent representation $\mu_{\hat{\boldsymbol{x}}}$, which helps maintain ranking consistency between original and reconstructed samples. To enforce correct survival risk orderings, DP-SurVAE introduces two partial ranking-based losses: $\mathcal{L}_{\text{lpl}}$ for enforcing correct risk orderings between original samples, and $\mathcal{L}_{\text{dplpl}}$ for maintaining these orderings under reconstruction and masking. The model is trained using the ELBO objective, reconstruction loss, mask sparsity regularization $\mathcal{L}_{\text{mask}}$, and the two ranking losses. The method is described in more detail below, where $\Theta$ refers to the set of all trainable parts and parameters of the proposed framework.

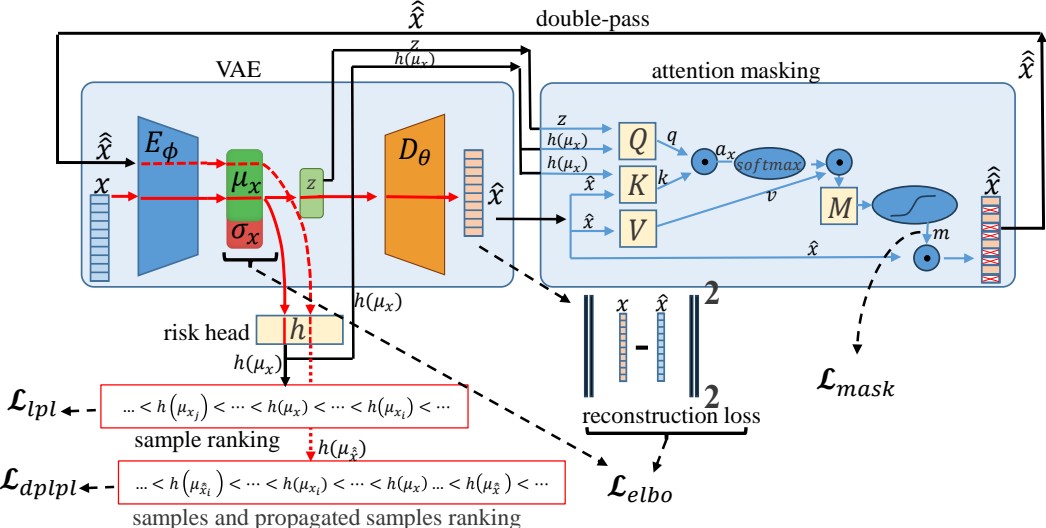

Figure 1: DP-SurVAE architecture. The Double-Pass Survival VAE (DP-SurVAE) consists of an encoder-decoder structure for variational inference, combined with a risk prediction head $h(\mu_{\boldsymbol{x}})$ applied to the latent mean. It incorporates an attention module (right box) to learn sparse feature masks guided by a quantile and sample-specific threshold.

## 4.1 VAE EMBEDDING

A VAE (Kingma et al., 2013) is a generative model that maps each input data point to a distribution in a latent space. A decoder then maps samples from this latent distribution back to the input space. The VAE consists of an *encoder* $E_\phi$ and a *decoder* $D_\theta$, parameterized by $\theta$ and $\phi$, respectively. These components are trained jointly by maximizing the Evidence Lower Bound (ELBO):

$$\mathcal{L}_{\text{elbo}}(\boldsymbol{x}; \Theta) = \mathbb{E}_{q_\phi(z|\boldsymbol{x})}[\log p_\phi(\boldsymbol{x}|z)] - \text{KL}(q_\phi(z|\boldsymbol{x}) \,||\, p(z)),$$

where $q_\phi(z|\boldsymbol{x})$ is the approximate posterior, modeled by the encoder; $p_\theta(\boldsymbol{x}|z)$ is the likelihood of the data given the latent variable, modeled by the decoder; $p(z)$ is the prior distribution over latent variables, typically $\mathcal{N}(0, I)$; and $\text{KL}(\cdot||\cdot)$ is the Kullback-Leibler divergence.

The first term is the *reconstruction loss*, encouraging the decoder to accurately reconstruct the input from the latent code. The second term is a *regularization loss*, ensuring that the learned posterior $q_\phi(z|\boldsymbol{x})$ stays close to the prior $p(z)$.

Typically, for the sample $\boldsymbol{x}$, the encoder's output models a Gaussian distribution in the latent space $E_\phi(\boldsymbol{x}) = \mu_x, log\sigma_x^2$, and the reparameterization trick is used to obtain the embedding $z_x = Gauss(\mu_x, \sigma_x^2)$, where $z_x$, $\mu_x$, $\sigma_x \in \mathbb{R}^l$ with $l$ being the size of the latent space. Finally, the VAE's output, $\hat{\boldsymbol{x}}$, is reconstructed $\hat{\boldsymbol{x}} = D_\theta(z_x)$.

## 4.2 RISK PREDICTION

Following the latent embedding, we retain the PH assumption and use a linear layer to model the log-risk function. Consequently, the time-independent component of the risk function in Eq. (1) is modeled as a non-linear transformation $r(\boldsymbol{x}) = \exp(h(\mu_{\boldsymbol{x}}))$, where $\mu_{\boldsymbol{x}}$ denotes the mean of the posterior distribution (used instead of a sampled latent vector $z$), and $h$ is a learnable function, typically a linear layer.

The function $h$ is trained by minimizing the negative Log-Partial Likelihood (LPL) in Eq. (5), which encourages correct event-time ranking despite the presence of censoring. For an uncensored sample $(\boldsymbol{x}, t, \delta)$, the likelihood is:

$$\mathcal{L}_{\text{lpl}}(\boldsymbol{x}; \Theta) = h(\mu_{\boldsymbol{x}}) - \log \sum_{(\boldsymbol{x}_j, t_j, \delta_j) \in R(t)} \exp(h(\mu_{\boldsymbol{x}_j})),$$

where $R(t) = \{(\boldsymbol{x}_j, t_j, \delta_j) \in D \mid t_j \geq t\}$ is the risk set.

### 4.3 MASKING

To enable sample-specific feature selection, we apply an attention-based masking mechanism that modulates the reconstructed output $\hat{\boldsymbol{x}}$.

Let $\boldsymbol{Q} \in \mathbb{R}^{l \times d}$, $\boldsymbol{K} \in \mathbb{R}^{(d+1) \times d}$, and $\boldsymbol{V} \in \mathbb{R}^{d \times d}$ denote the query, key, and value matrices, respectively. Let $M : \mathbb{R}^d \to \mathbb{R}^d$ denote the masking head, which produces a sample-specific mask. As depicted in Figure1, we compute the attention:

$$\boldsymbol{a}_x = \mathrm{softmax}(\boldsymbol{q} \odot \boldsymbol{k}^\top) \odot \boldsymbol{v} \in \mathbb{R}^d,$$

where $\odot$ is the Hadamard product. The mask is then generated from the attended features by applying a sigmoid activation to the masking head's output: $\boldsymbol{m} = \sigma(M(\boldsymbol{a}_x))$.

To promote sparsity, we sharpen the mask using a quantile-based, sample-specific threshold:

$$\tau = \mathrm{quantile}(\boldsymbol{m}, \alpha), \qquad \boldsymbol{m}' = \sigma\left(\beta \cdot (\boldsymbol{m} - \tau)\right)$$

where $\alpha \in (0, 1]$ is the target sparsity level, and $\beta \gg 1$ is a sharpening factor. The final masked reconstruction is obtained via element-wise multiplication: $\hat{\hat{\boldsymbol{x}}} = \boldsymbol{m}' \odot \hat{\boldsymbol{x}}$. To promote sparsity and interpretability in the mask, we define a masking loss composed of three regularization terms: $\ell_1$ sparsity, entropy regularization, and a budget constraint:

$$\mathcal{L}_{\mathrm{mask}}(\boldsymbol{x}; \Theta) = \lambda_1 \cdot \underbrace{\mathbb{E}[|\boldsymbol{m}|]}_{\text{L1 sparsity}} + \lambda_2 \cdot \underbrace{\mathbb{E}\left[H_b(\boldsymbol{m})\right]}_{\text{Entropy regularization}} + \lambda_3 \cdot \underbrace{(\mathbb{E}[\boldsymbol{m}] - \alpha)^2}_{\text{Budget penalty}}, \tag{3}$$

where the binary entropy function is defined as $H_b(p) = -p \log(p + \varepsilon) - (1 - p) \log(1 - p + \varepsilon)$, applied element-wise, and $\varepsilon$ is a small stability constant.

### 4.4 DOUBLE-PASS RISK PREDICTION

After the masking step, the masked decoded output $\hat{\hat{\boldsymbol{x}}}$ is re-encoded through the encoder (a process we refer to as *double-passing*), yielding a new log-risk prediction $h(\mu_{\hat{\hat{\boldsymbol{x}}}})$. For an uncensored sample $(\boldsymbol{x}, t, \delta)$, the *Double-Pass Log-Partial Likelihood* (DPLPL) takes the form

$$\mathcal{L}_{\mathrm{dplpl}}(\boldsymbol{x}; \Theta) = h(\mu_{\boldsymbol{x}}) + h(\mu_{\hat{\hat{\boldsymbol{x}}}}) - 2 \log \sum_{(\boldsymbol{x}_j, t_j, \delta_j) \in R} \left( e^{h(\mu_{\boldsymbol{x}_j})} + e^{h(\mu_{\hat{\hat{\boldsymbol{x}}}_j})} \right), \tag{4}$$

as derived in Appendix B.1. We additionally regularize the latent distributions induced by the encoder on the original and masked inputs, which is $E_\phi(\boldsymbol{x}) = (\mu_{\boldsymbol{x}}, \log \sigma_{\boldsymbol{x}}^2)$ and $E_\phi(\hat{\hat{\boldsymbol{x}}}) = (\mu_{\hat{\hat{\boldsymbol{x}}}}, \log \sigma_{\hat{\hat{\boldsymbol{x}}}}^2)$, by minimizing the KL divergence. This promotes consistency in the latent space across the original and double-passed inputs.

The need for the double-pass likelihood arises from the degradation of risk consistency caused by the masking mechanism. This degradation is formally derived in Theorem 1, which compares the consistency of risk predictions when masking is active versus inactive.

**Theorem 1** (Double-Pass Risk Consistency for Strong Masking)**.** *Assume the encoder $E_\phi$ and the risk head $h$ are $l_E$- and $l_h$-Lipschitz, respectively. Let $\hat{\boldsymbol{x}} = D(E(\boldsymbol{x}))$ be the VAE reconstruction, and $\hat{\hat{\boldsymbol{x}}}$ the double-masked version (with mask sharpness $\beta$ and sparsity $\alpha$).*

***Case 1 (No masking).*** *When masking is **not** applied (i.e., $\hat{\hat{\boldsymbol{x}}} = \hat{\boldsymbol{x}}$), the risk difference is bounded by a constant $\delta$: $|r(\boldsymbol{x}) - r(\hat{\hat{\boldsymbol{x}}})| \leq \delta$.*

***Case 2 (With masking).*** *When masking is applied (so $\hat{\hat{\boldsymbol{x}}} \neq \hat{\boldsymbol{x}}$), the bound becomes a function of the sparsity level $\alpha$: $|r(\boldsymbol{x}) - r(\hat{\hat{\boldsymbol{x}}})| \leq \delta(\alpha)$. However, this bound becomes looser with aggressive sparsification, degrading the consistency unless the model is explicitly trained to minimize this deviation.*

The consistency of the risk is an enforced property of the model through training, not a result of encoder and risk head continuity. Even under aggressive masking (e.g., $\alpha = 0.99$), the model preserves risk prediction fidelity via training, supporting robustness and interpretability of explanations. The proof of Theorem 1 can be found in Appendix B.2.

Maximizing Eq. equation 4 and aligning latent distributions achieve the objectives: *(i)* The ranking between masked samples aligns with the ground-truth order of events, and the ranking after masking remains similar to that before. *(ii)* The encoder and masking mechanism produce consistent embeddings that do not degrade the performance of the risk predictor.

Table 1: C-index comparison on miRNA cohorts; best value among the last five methods (DP-SurVAE, SurvSHAP, SurvLIME, SHAP, LIME) is in boldface. Standard errors (2nd row), ranks and one-sided Wilcoxon p-values vs. DP-SurVAE (last rows). The SurvVAE column reports the performance prior to applying the double-pass procedure.

Table 2: C-index comparison on mRNA cohorts; best value among the last four methods (DP-SurVAE, SurvLIME, SHAP, LIME) is highlighted in bold. Standard errors (2nd row), ranks and one-sided Wilcoxon p-values vs. DP-SurVAE (last rows). The SurvVAE column reports the performance prior to applying the double-pass procedure.

**Table 1**

| Cancer | # Selected Features | SurVAE | DP-SurVAE (Ours) | SurvSHAP | SurvLIME | SHAP | LIME |
|---|---|---|---|---|---|---|---|
| BLCA | 59.0 | .665 | **.661** | .542 | .537 | .576 | .534 |
|  | 5.2 | .02 | .03 | .03 | .03 | .02 | .04 |
| BRCA | 55.9 | .533 | .536 | .510 | **.547** | .476 | .481 |
|  | 3.4 | .03 | .03 | .04 | .04 | .05 | .05 |
| CESC | 56.8 | .715 | **.696** | .565 | .570 | .633 | .552 |
|  | 5.9 | .04 | .04 | .04 | .04 | .04 | .04 |
| COAD | 60.8 | .609 | **.613** | .497 | .533 | .504 | .552 |
|  | 4.6 | .03 | .04 | .06 | .05 | .05 | .05 |
| ESCA | 61.2 | .657 | **.652** | .530 | .580 | .573 | .607 |
|  | 5.9 | .02 | .03 | .03 | .05 | .03 | .04 |
| HNSC | 55.5 | .579 | **.579** | .521 | .516 | .548 | .529 |
|  | 5.9 | .03 | .03 | .02 | .02 | .03 | .03 |
| KIRC | 51.1 | .711 | **.713** | .533 | .517 | .636 | .543 |
|  | 5.7 | .01 | .01 | .04 | .04 | .03 | .03 |
| KIRP | 51.0 | .769 | **.763** | .595 | .620 | .731 | .691 |
|  | 4.9 | .05 | .04 | .05 | .08 | .06 | .06 |
| LGG | 58.3 | .746 | **.745** | .552 | .612 | .668 | .613 |
|  | 5.0 | .01 | .01 | .06 | .05 | .04 | .04 |
| LIHC | 61.0 | .543 | .538 | .486 | .526 | **.552** | .524 |
|  | 6.1 | .03 | .03 | .05 | .04 | .04 | .04 |
| LUAD | 50.9 | .600 | .593 | .546 | .521 | **.612** | .587 |
|  | 7.7 | .01 | .02 | .04 | .04 | .03 | .03 |
| LUSC | 58.7 | .546 | .544 | .527 | .476 | **.553** | .526 |
|  | 7.2 | .03 | .03 | .03 | .03 | .03 | .02 |
| MESO | 71.4 | .637 | .573 | .546 | .478 | **.605** | .577 |
|  | 4.2 | .03 | .05 | .06 | .04 | .05 | .06 |
| OV | 56.8 | .554 | **.535** | .512 | .500 | .519 | .482 |
|  | 10. | .03 | .03 | .02 | .02 | .02 | .02 |
| PAAD | 61.8 | .571 | **.567** | .546 | .502 | .519 | .549 |
|  | 7.4 | .05 | .05 | .05 | .04 | .06 | .05 |
| SARC | 55.7 | .609 | **.621** | .514 | .477 | .604 | .563 |
|  | 7.4 | .04 | .04 | .04 | .04 | .04 | .04 |
| STAD | 53.4 | .586 | **.588** | .532 | .514 | .523 | .527 |
|  | 5.9 | .02 | .02 | .03 | .03 | .03 | .02 |
| UCEC | 61.1 | .642 | **.629** | .533 | .617 | .505 | .496 |
|  | 4.0 | .04 | .03 | .05 | .07 | .04 | .04 |
| P-value |  |  |  | 7e-37 | 1e-33 | 1e-21 | 2e-31 |
| Rank |  |  | **1.33** | 3.83 | 3.94 | 2.5 | 3.39 |

**Table 2**

| Cancer | # Selected Features | SurVAE | DP-SurVAE (Ours) | SurvLIME | SHAP | LIME |
|---|---|---|---|---|---|---|
| BLCA | 911 | .703 | **.702** | .631 | .651 | .666 |
|  | 261 | .01 | .02 | .02 | .02 | .01 |
| BRCA | 914 | .744 | **.744** | .637 | .691 | .664 |
|  | 263 | .03 | .03 | .03 | .03 | .03 |
| CESC | 654 | .714 | **.686** | .640 | .675 | .649 |
|  | 181 | .03 | .04 | .04 | .03 | .03 |
| COAD | 686 | .661 | **.650** | .591 | .639 | .615 |
|  | 183 | .04 | .03 | .05 | .04 | .05 |
| ESCA | 485 | .531 | **.519** | .517 | .517 | .508 |
|  | 157 | .03 | .03 | .04 | .04 | .04 |
| GBM | 357 | .625 | .579 | .590 | **.629** | .594 |
|  | 113 | .02 | .04 | .02 | .03 | .03 |
| HNSC | 749 | .648 | **.649** | .602 | .633 | .625 |
|  | 214 | .02 | .02 | .02 | .02 | .02 |
| KIRC | 815 | .691 | **.690** | .577 | .660 | .628 |
|  | 269 | .02 | .02 | .03 | .03 | .03 |
| KIRP | 682 | .759 | .735 | **.754** | .711 | .698 |
|  | 161 | .07 | .08 | .06 | .07 | .06 |
| LGG | 901 | .824 | **.825** | .753 | .823 | .791 |
|  | 231 | .03 | .03 | .03 | .03 | .03 |
| LIHC | 865 | .587 | .598 | .524 | **.603** | .591 |
|  | 233 | .02 | .03 | .03 | .04 | .05 |
| LUAD | 711 | .626 | .630 | .568 | .630 | **.631** |
|  | 241 | .03 | .03 | .04 | .03 | .03 |
| LUSC | 994 | .534 | .531 | .541 | .540 | **.543** |
|  | 244 | .02 | .02 | .02 | .02 | .03 |
| MESO | 349 | .713 | .646 | .618 | **.652** | .621 |
|  | 74 | .03 | .04 | .06 | .05 | .05 |
| OV | 669 | .614 | **.612** | .527 | .586 | .565 |
|  | 187 | .01 | .01 | .02 | .02 | .02 |
| PAAD | 429 | .720 | **.699** | .541 | .643 | .641 |
|  | 107 | .02 | .02 | .05 | .05 | .03 |
| SARC | 643 | .690 | **.681** | .629 | .652 | .649 |
|  | 212 | .04 | .03 | .04 | .03 | .03 |
| STAD | 681 | .615 | .608 | **.622** | .582 | .577 |
|  | 281 | .03 | .03 | .03 | .03 | .05 |
| UCEC | 867 | .581 | **.568** | .538 | .541 | .529 |
|  | 240 | .02 | .02 | .04 | .06 | .04 |
| P-value |  |  | 8.5e-29 | 1.4e-8 | 1.9e-19 | 2.8e-36 |
| Rank |  |  | **1,63** | 3,42 | 2,05 | 2,89 |

## 4.5 FINAL OBJECTIVE

The method tries to find the trainable parts, $\Theta$, by minimizing the total loss $\mathcal{L}_{\text{total}}$, which leads to maximizing the ELBO and the two partial likelihoods while minimizing the mask loss:

$$\mathcal{L}_{\text{total}} = -\mathcal{L}_{elbo} - \mathcal{L}_{lpl} + \mathcal{L}_{\text{mask}} - \mathcal{L}_{\text{dplpl}}.$$

# 5 EMPIRICAL EVALUATION

## 5.1 USED DATASET

The data used in this work are mRNA and miRNA from cancer patients, generated by the TCGA Research Network.[1] Detailed information on cancer types (also referred to as cohorts), the number of patients and events (i.e., deaths) are listed in Table 3 and Table 4 in the supplementary material. Section C describes the adopted criteria for inclusion or exclusion of cohorts. Data acquisition and processing are described in Appendices C.1 and C.2, respectively.

## 5.2 PRACTICAL DETAILS

We follow the work of InfoVAE (Zhao et al., 2019), where it is shown that the KL divergence often leads to latent space collapse to the prior. Instead, they suggest replacing the KL divergence with the Maximum Mean Discrepancy (MMD) (Gretton et al., 2006). As for the parametrizations, we use $l = 20$ for the latent space, and for the desired sparsity level $\alpha$ (0.99 for miRNA and 0.999 for mRNA), $\beta = 10^3$, and $\lambda_1 = \lambda_2 = \lambda_3 = 1$ for the weights of the mask regularization. In practice, and for high-dimensional data, we do not use attention matrices for $\boldsymbol{K}, \boldsymbol{V}$, and $\boldsymbol{M}$. Instead, we use a low-rank (bottleneck) approximation followed by a nonlinearity and reconstruction. The bottleneck size is 20.

## 5.3 BASELINES

We compare our method on both TCGA datasets (mRNA and miRNA) against SurvLIME, SurvSHAP, SHAP, and LIME. Since both SurvLIME and SurvSHAP operate as explainers for survival functions, and since our method estimates the proportional time-independent hazard function $\lambda(\boldsymbol{x})$ in Eq. (1), we apply SurvLIME and SurvSHAP to $S(t; \boldsymbol{x}) = e^{-H(t; \boldsymbol{x})}$, where $H(t; \boldsymbol{x}) = \int_0^t h(u; \boldsymbol{x}) \, du$ is the cumulative hazard function and $t$ is taken from the interval covering the lifespan of all patients. SHAP and LIME, on the other hand, are applied directly to the predicted risks.

## 5.4 RESULTS

For a fair comparison between DP-SurVAE and the other methods, we take the average maximum number of activated features, $K$, found by DP-SurVAE and instruct each method to discover the $K$ most relevant features. The data is then projected onto these features, and the model is evaluated on the projected data. Moreover, for more baselines, we use the projected data from each method to train and test a DeepSurv (Katzman et al., 2018) model. SurvSHAP is skipped on the mRNA data, since one fold takes 31 hours, implying 123 days of compute time for the full mRNA—which exceeded our time and compute budget. Results are presented in terms of the C-index, see Section D.1.

Table 1 reports the C-index for all feature attribution methods across the different miRNA cancer cohorts; Table 5 shows the C-index of their corresponding DeepSurv (DS). The number of selected features lies between roughly 2.7% and 3.7% of the full set of 1 911 features, reflecting significant dimensionality reduction achieved by DP-SurVAE. Moreover, it consistently achieves the highest or near-highest C-index in the majority of cohorts, demonstrating robust predictive accuracy. It ranks first on average (1.33), substantially outperforming SurvSHAP, SurvLIME, SHAP, and LIME, as well as their DeepSurv-trained variants. While the DeepSurv baselines (DS variants) sometimes reach competitive performance, they generally fall short of DP-SurVAE. For a few cohorts, such as KIRP and LIHC, DS-SHAP or DS-SurvSHAP achieve strong results, but DP-SurVAE still retains the top overall position.

Similarly, Tables 2 and 6 shows the C-index across the mRNA cohorts. The number of selected features amounts to roughly 0.89% to 2.5% of the total 39 205 features, highlighting a considerably better dimensionality reduction. Our method, DP-SurVAE, performs well on the majority of mRNA cohorts and achieves the best average rank (1,63). Although DS-SHAP comes out on top in a few individual cases—such as CESC, LIHC, and MESO, DP-SurVAE remains consistently among the best-performing approaches overall.

---

[1]https://www.cancer.gov/tcga

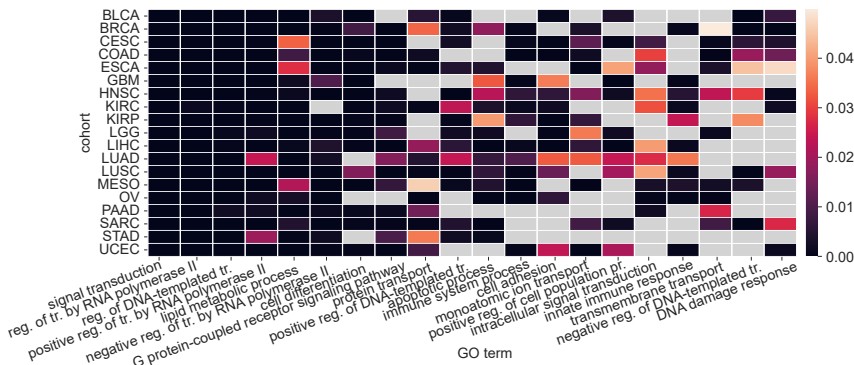

Figure 2: The x-axis shows all enriched GO terms that occurred at least in 10 different cohorts for DP-SurVAE. On the y-axis, all cohorts are listed. The color on the heatmap indicates the p-values corrected by false discovery rate resampling. Some terms were abbreviated: Regulation (reg.), transcriptions (tr.), proliferation (pr.)

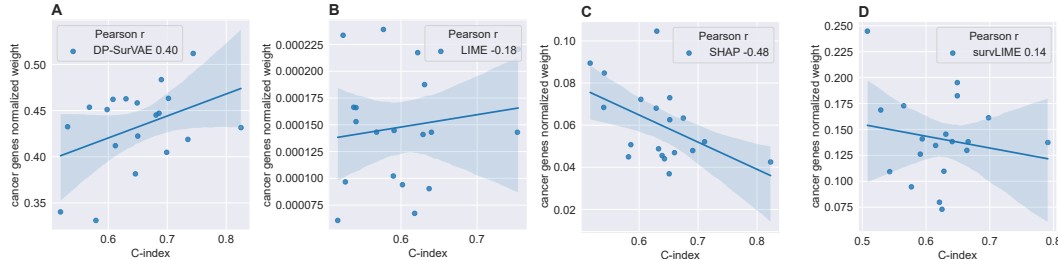

Figure 3: Correlation between C-indices and the mean normalized weight of cancer genes among the top 3 000 most important genes. Each dot represents one cohort. A: our method, DP-SurVAE; B: LIME; C: SHAP; D: SurvLIME.

Compared to SurvSHAP, SurvLIME, SHAP, and LIME, as well as their DeepSurv-trained variants, DP-SurVAE demonstrates superior results on both datasets, which are also statistically significant, with p-values $\leq 1.3 \times 10^{-3}$ in all cases based on the one-sided Wilcoxon signed-rank test, indicating that the observed gains are unlikely to be due to chance.

## 6 BIOLOGICAL RELEVANCE OF FEATURES

After feature selection, whether obtained from DP-SurVAE or from post-hoc baselines, we directly use the features for biological evaluation. Specifically, we assess faithfulness by examining whether the identified features correspond to biological processes relevant for tumor formation and progression, which has direct implications on patients' survival. We evaluate the selected mRNA features from two perspectives. (1) We perform Gene Ontology (GO) analysis to test for each method whether these sample-wise selected features are enriched in terms related to biological processes. (2) We perform correlation analysis between the performance of each method and its identified feature importances of known cancer genes.

### 6.1 ENRICHMENT OF GENE ONTOLOGY TERMS

GO terms are standardized terms to describe and classify the functions of genes. GO term enrichment analysis aims to investigate whether terms[2] associated with genes are statistically enriched in a selected gene set relative to a reference set (Ashburner et al., 2000). We select the top 3 000 mRNA features, genes, with the highest mask weight per cohort and perform an enrichment analysis using the GOATOOLS method (Klopfenstein et al., 2018). The gene reference set comprises all 45 671

---

[2]https://geneontology.org/, accessed 28.7.25

annotated genes in NCBI.[3] In total, we analyzed 18 774 GO terms from the "biological processes" category and identified 393 terms for DP-SurVAE that are statistically enriched in at least one cancer cohort.

Figure 2 shows a heatmap of the p-values, corrected by false discovery rate resampling, for all terms occurring in at least 10 different cohorts for DP-SurVAE. Many enriched terms refer to processes allowing tumor growth, such as terms related to RNA processing, thus affecting the regulation of gene expression and cell proliferation (Hanahan, 2022; Obeng et al., 2019), as well as DNA processing, which contributes to genome instability and the accumulation of mutations (Hanahan, 2022). Immune system terms are also present, since tumors modulate their interaction with the immune system to evade detection (Hanahan, 2022; Gupta et al., 2023). Another prominent family of terms refers to signaling, which is connected to many hallmarks of cancer, allowing the tumor to control cellular processes and its environment to survive and grow (Sever & Brugge, 2015). We also looked at the 20 terms with the highest corrected p-value mean over all cohorts. While all terms for DP-SurVAE are related to cell or metabolic functions necessary for tumors (Table 9), there are some terms for the other methods, which appear to be false positives, such as "nervous system development" for SurvLIME (Table 10) and "antimicrobial humoral immune response mediated by antimicrobial peptide" for SHAP (Table 11) and LIME (Table 12).

## 6.2 CANCER GENE ANALYSIS

We use the list of known cancer genes from OncoKB[4] (Chakravarty et al., 2017; Suehnholz et al., 2024) and match the gene names with those from HGNC, as described in Section C.1, to ensure that no genes are missed due to naming inconsistencies. The list contains 1 192 genes, of which 979 are found in at least one cohort. Each cohort is represented by the 3 000 most highly weighted genes, with weights normalized for comparison across cohorts. Taking the median of the normalized weights for cancer and non-cancer genes, we observe that in 12 out of 21 cohorts, the cancer genes' medians surpass those of the non-cancer genes. Figure 3 shows that a higher mean of the normalized weights correlates with an increased C-index for our method (0.40 Pearson correlation) while no or no positive correlation can be observed for the other methods (-0.18, -0.48 and 0.14 for LIME, SHAP and survLIME, respectively). A higher C-index means better performance, which is supported by a higher masking weight, that is, feature importance, of cancer genes. A positive correlation shows that our method considers known cancer genes more often for cohorts for which it achieves better performance. This implies that for these cohorts the method learns the underlying biological relevance of the selected genes better. The selected genes and their expression profile can be converted into gene expression signatures (Giampietri et al., 2022).

These findings increase confidence in the ability of our method to detect novel gene expression signatures for diagnostic or treatment purposes. Traditionally, these signatures are established by considering known genes only (Giampietri et al., 2022) but it has already been shown that machine learning approaches are capable of detecting novel signatures (Tschodu et al., 2023). However, previous studies focus on very narrowly defined subsets of cancer patients and ignore the censoring problem, which our method addresses (Giampietri et al., 2022; Tschodu et al., 2023).

## 7 CONCLUSION

In this work, we presented a novel approach that equips survival analysis with interpretable feature importance and provides risk consistency guarantees. Our method, DP-SurVAE, introduces a second pass through the VAE (double-pass) and optimizes a novel double-pass log-partial likelihood objective, which explicitly accounts for the event partial ranking between censored and uncensored samples. Empirical and biological evaluations on the TCGA dataset demonstrate that DP-SurVAE achieves superior C-index performance compared to existing methods. Moreover, the model more effectively identifies biologically relevant genes, enhancing the interpretability of risk predictions. The confirmation of biological relevance increases confidence in our method and its prediction ability, as well as in the potential to use its masking weight to identify gene expression signatures for diagnostic and therapeutic purposes.

---

[3]https://www.ncbi.nlm.nih.gov/gene, downloaded 7.5.25
[4]https://www.oncokb.org/cancer-genes, downloaded 11.6.25

## 8 ETHICS STATEMENT

Our data analysis procedures strictly follow the GDPR regulations for collecting and processing personal data, as detailed in Data Compliance with GDPR[5]. While the TCGA[6] dataset is already anonymized, we perform an additional anonymization step to further ensure that no link can be established between the original samples and the inputs used in our analyses.

## 9 REPRODUCIBILITY STATEMENT

For reproducibility, we provide our code in an anonymous Git repository. The implementation files are shared under an anonymous license and may be used for reviewing purposes only[7]. The data anonymization procedure is included in the shared implementation. Details of the parametrizations used in our method and the baselines are provided in Appendix G. The proof of Theorem 1 is given in Appendix B.2, and the derivation of the Double-Pass Partial Likelihood is presented in Appendix B.1.

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

# Supplementary Material

## Survival VAE:
## Robust Local Explanations via Double-Pass Risk Consistency

### Anonymous Submission

## A  RELATED WORK: MACHINE LEARNING FOR SURVIVAL ANALYSIS

The Kaplan–Meier (KM) estimator (Kaplan & Meier, 1958) is a nonparametric method that estimates the survival function without assuming any underlying distribution and is effective for moderate-sized datasets. The CPH model addresses this limitation by assuming proportional hazards between individuals and modeling the log-risk as a linear function of the covariates. Various machine learning techniques have been adapted to survival data. Tree-based methods have played a central role in survival analysis, offering flexible, nonparametric approaches for censored data. Early work adapted classical decision trees to the survival setting using splitting rules based on log-rank test Segal (1988). Random Survival Forests (RSF) (Ishwaran & Kogalur, 2007) leverage ensemble learning to handle censoring, Bayesian additive regression trees (Survival BART) offer a probabilistic framework for nonparametric survival modeling Sparapani et al. (2016). Support Vector Regression for Censored data (SVRC) (Khan & Zubek, 2008) adapts support vector machines to the survival setting; see also the comprehensive survey by Wang et al. (2019).

Tremendous advances in deep learning have led to significant progress in survival analysis. DeepSurv leverages a neural network trained with a loss aligned to the PH assumption, retaining hazard proportionality while replacing the log-linear risk with a more expressive nonlinear model—resulting in strong performance gains. In contrast, DeepHit (Lee et al., 2018) relaxes the PH assumption by learning time-varying survival distributions and supporting competing risks and multiple events per patient. Liu et al. (2024) apply copulas to model dependent censoring with identifiability guarantees. Building on hidden mathematical structures of the Cox model. Zhang et al. (2024) propose an efficient optimization procedure that fits the Cox model by minimizing surrogate functions. Shaker & Lawrence (2023) introduce a multi-source domain adaptation framework for survival analysis.

## B  MATHEMATICAL PROOFS

### B.1  DOUBLE-PASS PARTIAL LIKELIHOOD

For a dataset with $m$ uncensored events occurring at distinct times, the partial likelihood is given by the product of the individual partial likelihoods for each event:

$$PL(\Theta) = \prod_{o=1}^{m} L(\boldsymbol{x}_o; \Theta) \ .$$

For an uncensored sample $(\boldsymbol{x}, t, \delta)$, the log-risk is given by $h(\mu_{\boldsymbol{x}})$ for the original sample, and by $h(\mu_{\hat{\boldsymbol{x}}})$ for the same sample after masking $\boldsymbol{x}$ (denoted $\hat{\boldsymbol{x}}$) and applying the double-pass. The conditional probability for an event occurring at time $t$ takes into account the relative ranking of

both $h(\mu_{\boldsymbol{x}})$ and $h(\mu_{\hat{\boldsymbol{x}}})$ versus the ranking of all other samples in the extended risk set:

$$
\begin{aligned}
L(\boldsymbol{x};\Theta) &= \frac{e^{h(\mu_{\boldsymbol{x}})}}{\sum_{(\boldsymbol{x}_j,t_j,\delta_j)\in R}\left(e^{h(\mu_{\boldsymbol{x}_j})}+e^{h(\mu_{\hat{\boldsymbol{x}}_j})}\right)} \\
&\quad \cdot \frac{e^{h(\mu_{\hat{\boldsymbol{x}}})}}{\sum_{(\boldsymbol{x}_j,t_j,\delta_j)\in R}\left(e^{h(\mu_{\boldsymbol{x}_j})}+e^{h(\mu_{\hat{\boldsymbol{x}}_j})}\right)} \\
&= \frac{e^{h(\mu_{\boldsymbol{x}})}\cdot e^{h(\mu_{\hat{\boldsymbol{x}}})}}{\left(\sum_{(\boldsymbol{x}_j,t_j,\delta_j)\in R}\left(e^{h(\mu_{\boldsymbol{x}_j})}+e^{h(\mu_{\hat{\boldsymbol{x}}_j})}\right)\right)^2}\ ,
\end{aligned}
\tag{5}
$$

where $R(t) = \{(\boldsymbol{x}_l,t_l,\delta_l)\in D \mid t_l \geq t\} \cup \{(\hat{\boldsymbol{x}}_l,t_l,\delta_l)\in D \mid t_l \geq t\}$ is the extended risk set, which includes each sample twice — once before and once after masking — provided its survival time is greater than or equal to that of the sample $\boldsymbol{x}$ under consideration.

Therefore, *Double-Pass Log-Partial Likelihood* (DPLPL) for $\boldsymbol{x}$ becomes:

$$
\mathcal{L}_{\text{dplpl}}(\boldsymbol{x};\Theta) = h(\mu_{\boldsymbol{x}}) + h(\mu_{\hat{\boldsymbol{x}}}) - 2\log\sum_{(\boldsymbol{x}_j,t_j,\delta_j)\in R}\left(e^{h(\mu_{\boldsymbol{x}_j})}+e^{h(\mu_{\hat{\boldsymbol{x}}_j})}\right)
\tag{6}
$$

## B.2 Double-Pass Risk Consistency

The following theorem demonstrates that the risk consistency is not derived from continuity assumptions alone, due to the discontinuous behavior of $\|\hat{\hat{\boldsymbol{x}}} - \hat{\boldsymbol{x}}\|$ under masking. Instead, consistency emerges from explicit optimization of double-pass likelihood objective $\mathcal{L}_{\text{dplpl}}$. This explains the robustness of the risk prediction even under coarse masking, where a large portion of features is dropped.

**Theorem 1** (Double-Pass Risk Consistency for Strong Masking). *Assume the encoder $E_\phi$ and the risk head $h$ are $l_E$- and $l_h$-Lipschitz, respectively. Let $\hat{\boldsymbol{x}} = D(E(\boldsymbol{x}))$ be the VAE reconstruction, and $\hat{\hat{\boldsymbol{x}}}$ the double-masked version (with mask sharpness $\beta$ and sparsity $\alpha$).*

- *When masking is **not** applied (i.e., $\hat{\hat{\boldsymbol{x}}} = \hat{\boldsymbol{x}}$), the risk difference is bounded by a constant $\delta$:*

$$
|r(\boldsymbol{x}) - r(\hat{\hat{\boldsymbol{x}}})| \leq \delta
$$

- *When masking is applied (so $\hat{\hat{\boldsymbol{x}}} \neq \hat{\boldsymbol{x}}$), the bound becomes a function of the sparsity level $\alpha$:*

$$
|r(\boldsymbol{x}) - r(\hat{\hat{\boldsymbol{x}}})| \leq \delta(\alpha)
$$

*However, this bound becomes looser with aggressive sparsification, degrading the consistency unless the model is explicitly trained to minimize this deviation.*

*Proof of Theorem 1.* In the following, we show the proof for the settings when the masking is active or not.

- **Case 1 (No Masking):** We list the Lipschitz continuity assumptions for inactive masking ($\hat{\boldsymbol{x}} = \hat{\hat{\boldsymbol{x}}}$):

  A1 Encoder Lipschitz continuity: The encoder $E_\phi$ is $l_E$-Lipschitz:

$$
\|E_\phi(\boldsymbol{x}_1) - E_\phi(\boldsymbol{x}_2)\| \leq l_E \cdot \|\boldsymbol{x}_1 - \boldsymbol{x}_2\|
$$

  A2 Risk head Lipschitz continuity: The risk head $h$ is $l_h$-Lipschitz:

$$
|h(\boldsymbol{x}_1) - h(\boldsymbol{x}_2)| \leq l_h \cdot \|\boldsymbol{x}_1 - \boldsymbol{x}_2\|
$$

  From the triangle inequality, we observe:

$$
\|\hat{\hat{\boldsymbol{x}}} - \boldsymbol{x}\| \leq \|\hat{\hat{\boldsymbol{x}}} - \hat{\boldsymbol{x}}\| + \|\hat{\boldsymbol{x}} - \boldsymbol{x}\| \leq \varepsilon_{\text{err}}\ ,
$$

where $\varepsilon_{\text{err}}$ is the VAE's reconstruction error, implemented as the squared $\ell_2$ norm. From assumption [A1], it follows that:

$$\|E_\phi(\hat{\hat{\boldsymbol{x}}}) - E_\phi(\boldsymbol{x})\| \le l_E \cdot \|\hat{\hat{\boldsymbol{x}}} - \boldsymbol{x}\| \le l_E \cdot \varepsilon_{\text{err}} \;.$$

Finally, applying the Lipschitz continuity of the risk head $h$ and assuming that $|h(E_\phi(\boldsymbol{x}))| \le C$ for any $\boldsymbol{x}$, we obtain:

$$\begin{aligned}
|r(\hat{\hat{\boldsymbol{x}}}) - r(\boldsymbol{x})| &= |\exp(h(\mu_{\hat{\boldsymbol{x}}})) - \exp(h(\mu_{\boldsymbol{x}}))| \\
&\le \exp(C) \cdot |h(\mu_{\hat{\boldsymbol{x}}}) - h(\mu_{\boldsymbol{x}})| \\
&\le \exp(C) \cdot l_h \cdot l_E \cdot \varepsilon_{\text{err}} = \delta
\end{aligned}$$

where the exponential function is $e^C$-Lipschitz continuous on any bounded interval $[-C, C]$.

- **Case 2 (With Masking):** When masking is applied, $\hat{\hat{\boldsymbol{x}}} \ne \hat{\boldsymbol{x}}$, and the deviation $\delta(\alpha, \beta) := \|\hat{\boldsymbol{x}} - \hat{\hat{\boldsymbol{x}}}\|$ is non-negligible and depends on target sparsity level $\alpha$ and sharpening factor $\beta$. While the two Lipschitz assumptions in Case 1 remain valid, the following changes:

$$\|\hat{\hat{\boldsymbol{x}}} - \boldsymbol{x}\| \le \|\hat{\hat{\boldsymbol{x}}} - \hat{\boldsymbol{x}}\| + \|\hat{\boldsymbol{x}} - \boldsymbol{x}\| \le \delta(\alpha, \beta) + \varepsilon_{\text{err}} \;.$$

It follows that:

$$\begin{aligned}
|r(\hat{\hat{\boldsymbol{x}}}) - r(\boldsymbol{x})| &= |\exp(h(\mu_{\hat{\boldsymbol{x}}})) - \exp(h(\mu_{\boldsymbol{x}}))| \\
&\le \exp(C) \cdot |h(\mu_{\hat{\boldsymbol{x}}}) - h(\mu_{\boldsymbol{x}})| \\
&\le \exp(C) \cdot l_h \cdot l_E \cdot (\varepsilon_{\text{err}} + \delta(\alpha, \beta)) \;.
\end{aligned}$$

This shows clearly that the risk consistency degrades when masking is activated and no countermeasure is used. This motivates the introduction of the loss term $\mathcal{L}_{\text{dplpl}}$, since it explicitly trains the model to enforce:

$$|h(\mu_{\hat{\boldsymbol{x}}}) - h(\mu_{\boldsymbol{x}})| \le \varepsilon(\alpha) \;,$$

ensuring that $\varepsilon(\alpha) \ll l_h \cdot l_E$ even when $\delta(\alpha, \beta)$ is large.

$\square$

## B.3 DISCUSSION ON ALTERNATIVE EVALUATION MEASURES

Besides, the c-index, there are metrics such as the Brier score. For survival analysis, this would mean evaluating a proper scoring rule such as the Continuous Ranked Probability Score (CRPS), which is the continuous analogue of the Brier score, while employing Inverse Probability of Censoring Weighting (IPCW): $\text{CRPS}_{\text{IPCW}} = \int_0^\infty w(t), (\hat{S}(t) - \mathbf{1}T > t)^2, dt$.

Having that said, the following highlights the obstacles and the potential bias that would be added if our risk predictor were evaluated using IPCW. Let us assume the function $f\colon X \to X$ that comprises our method DP-SurVAE except for the risk head, i.e., $f(x)$ maps the sample $x$ to the masked sample $\hat{x}$. The log-risk head $h$ takes any sample from $X$ and maps it to its risk. Hence, $h(f(x))$ maps the sample to its masked sample and then computes the risk. Every post-hoc method (SHAP, SurvSHAP, LIME, SurvLIME) tries to substitute $f(x)$ with a better feature-importance explanation function $g(x)$. The correctness of the ranking is evaluated using the C-index, as shown in the manuscript. In order to compute CRPS, however, we would need to compute the full survival curve: $\hat{S}(t) = \exp\left(-H_0(t)\exp(h(f(x)))\right)$, for DP-SurVAE, and $\hat{S}(t) = \exp\left(-H_0(t)\exp(h(g(x)))\right)$ for every other method. While the C-index depends only on the ranking, CRPS additionally depends on the baseline cumulative hazard $H_0(t)$, which requires extra steps involving survival-curve calibration that we do not cover in our solution. Importantly, the calibrated $H_0(t)$ is independent of the model components $h$, and $f$, as well as the baseline models $g$, and is therefore not part of the quantities we compare across approaches. An evaluation using CRPS/IBS would therefore not only evaluate our feature-importance strategy but also the calibration component, which is out of scope at this stage.

## C    DATA

The data used in this work are mRNA and miRNA from cancer patients, generated by the TCGA Research Network.[8] Detailed information on cancer types (also referred to as cohorts), the number of patients and events (i. e., deaths) are listed in Table 3 and 4. In both datasets, we include only cohorts with at least 25 events (deaths) and more than 80 samples; this results in 19 mRNA cohorts and 18 miRNA cohorts. Discrepancies between number of cohorts and samples per cohort are caused by differences in experiments done for each patient. The mRNA dataset contains 39 205 gene expression features, while the miRNA dataset includes 1 911 miRNA expression features.

### C.1    ACQUISITION

We downloaded manifest files from Genomic Data Commons[9] (GDC) by selecting cohorts from TCGA using the condition:

```
cases.project.program.name in ["TCGA"] and
files.access in ["open"] and
files.experimental_strategy
in ["RNA-Seq", "miRNA-Seq"]
and cases.samples.sample_type
in ["primary tumor"]
```

in the advanced search option of the gdc-client tool. Clinical annotation, which includes information about days from diagnosis to event or last follow-up, was downloaded in a similar way using the default repository search with data category `"clinical"`.

### C.2    DATA PROCESSING AND ANNOTATION

We normalize the raw gene expression data using:

$$y = \log(x + 1),$$

where adding 1 prevents taking the logarithm of zero. Furthermore, we link the clinical and gene expression data by retrieving and matching the file identifiers using the GDC API. To allow further analysis of the mRNA dataset, which relies on gene names, we map gene names to the HGNC-approved gene names, also called symbols, by using alias and previous symbols of the HUGO Gene Nomenclature Committee[10] (HGNC). Genes that could not be mapped were removed.

## D    ADDITIONAL RESULTS

### D.1    C-INDEX

The Concordance-index (C-index) (Harrell et al., 1982) measures how well the rank of predicted risks agrees with the order of the observed outcomes. To this end, for each uncensored instance, it counts how often the model ranks it at higher risk than instances who survive longer:

$$\text{C-index}(r; D) = \frac{1}{Z} \sum_{\substack{(\boldsymbol{x}_i, t_i, \delta_i) \in D \\ \wedge \delta_i = 1}} \sum_{\substack{(\boldsymbol{x}_j, t_j, \delta_j) \in D \\ \wedge t_j > t_i}} I[r(\boldsymbol{x}_i) > r(\boldsymbol{x}_j)], \tag{7}$$

where $Z$ is the total number of comparable pairs and $I[\cdot]$ is the indicator function. The function $r(\cdot)$ denotes the model's risk score. A higher C-index shows better agreement between predicted risks and actual survival times.

In terms of runtime, a direct comparison between DP-SurVAE and the other methods is challenging, as our approach includes both full model training and the additional cost of learning the explanatory

---

[8]https://www.cancer.gov/tcga

[9]https://portal.gdc.cancer.gov, downloaded 18.10.23

[10]https://www.genenames.org, downloaded 6.3.25

| ID | Cancer Acronym | Cancer Name | Primary Site | Instances (count) | $\delta = 1$ |
|----|----------------|-------------|--------------|-------------------|--------------|
| 1 | BLCA | Bladder Urothelial Carcinoma | Bladder | 412 | 112 |
| 2 | BRCA | Breast Invasive Carcinoma | Breast | 1110 | 104 |
| 3 | CESC | Cervical Squamous Cell Carcinoma | Cervix | 304 | 60 |
| 4 | COAD | Colon Adenocarcinoma | Colon | 479 | 56 |
| 5 | ESCA | Esophageal Carcinoma | Esophagus | 184 | 57 |
| 6 | GBM | Glioblastoma Multiforme | Brain | 156 | 103 |
| 7 | HNSC | Head and Neck Squamous Cell Carcinoma | Head and Neck | 520 | 167 |
| 8 | KIRC | Kidney Renal Clear Cell Carcinoma | Kidney | 541 | 160 |
| 9 | KIRP | Kidney Renal Papillary Cell Carcinoma | Kidney | 290 | 32 |
| 10 | LGG | Brain Lower Grade Glioma | Brain | 515 | 92 |
| 11 | LIHC | Liver Hepatocellular Carcinoma | Liver | 371 | 89 |
| 12 | LUAD | Lung Adenocarcinoma | Lung | 539 | 128 |
| 13 | LUSC | Lung Squamous Cell Carcinoma | Lung | 502 | 158 |
| 14 | MESO | Mesothelioma | Pleura | 87 | 58 |
| 15 | OV | Ovarian Serous Cystadenocarcinoma | Ovary | 421 | 234 |
| 16 | PAAD | Pancreatic Adenocarcinoma | Pancreas | 178 | 59 |
| 17 | SARC | Sarcoma | Soft Tissue | 259 | 75 |
| 18 | STAD | Stomach Adenocarcinoma | Stomach | 412 | 79 |
| 19 | UCEC | Uterine Corpus Endometrial Carcinoma | Uterus | 553 | 45 |

Table 3: Summary of the number of instances and events ($\delta = 1$) for each cancer type in the mRNA dataset, including their TCGA identifier (ID) and primary tumor site.

| ID | Cancer Acronym | Cancer Name | Primary Site | Instances (count) | $\delta = 1$ |
|----|----------------|-------------|--------------|-------------------|--------------|
| 1 | BLCA | Bladder Urothelial Carcinoma | Bladder | 417 | 112 |
| 2 | BRCA | Breast Invasive Carcinoma | Breast | 1095 | 102 |
| 3 | CESC | Cervical Squamous Cell Carcinoma | Cervix | 307 | 60 |
| 4 | COAD | Colon Adenocarcinoma | Colon | 453 | 55 |
| 5 | ESCA | Esophageal Carcinoma | Esophagus | 186 | 57 |
| 6 | HNSC | Head and Neck Squamous Cell Carcinoma | Head and Neck | 523 | 169 |
| 7 | KIRC | Kidney Renal Clear Cell Carcinoma | Kidney | 544 | 160 |
| 8 | KIRP | Kidney Renal Papillary Cell Carcinoma | Kidney | 291 | 32 |
| 9 | LGG | Brain Lower Grade Glioma | Brain | 511 | 91 |
| 10 | LIHC | Liver Hepatocellular Carcinoma | Liver | 372 | 88 |
| 11 | LUAD | Lung Adenocarcinoma | Lung | 519 | 125 |
| 12 | LUSC | Lung Squamous Cell Carcinoma | Lung | 478 | 147 |
| 13 | MESO | Mesothelioma | Pleura | 87 | 58 |
| 14 | OV | Ovarian Serous Cystadenocarcinoma | Ovary | 487 | 269 |
| 15 | PAAD | Pancreatic Adenocarcinoma | Pancreas | 178 | 59 |
| 16 | SARC | Sarcoma | Soft Tissue | 259 | 76 |
| 17 | STAD | Stomach Adenocarcinoma | Stomach | 446 | 85 |
| 18 | UCEC | Uterine Corpus Endometrial Carcinoma | Uterus | 545 | 44 |

Table 4: Summary of the number of instances and events ($\delta = 1$) for each cancer type in the miRNA dataset, including their TCGA identifier (ID) and primary tumor site.

components (e.g., double-pass and masking). Despite this, DP-SurVAE—including model training—consistently outperforms all other methods in terms of speed, with the exception of SHAP. Tables 7 and 8 present the runtimes in seconds.

We additionally considered the 20 terms with the highest corrected p-value mean across all cohorts. For DP-SurVAE, all terms are related to cell or metabolic processes essential for tumors (Table 9). In contrast, some terms from the other methods appear to be false positives, such as "nervous system development" for SurvLIME (Table 10) and "antimicrobial humoral immune response mediated by antimicrobial peptide" for SHAP (Table 11) and LIME (Table 12).

Below, we present a list of Tables that depict the runtime on the mRNA and miRNA data, and the enriched GO terms of the different methods on the mRNA data.

1. Table 3 summarizes the number of instances and events ($\delta = 1$) for each cancer type in the mRNA dataset, including their TCGA identifier (ID) and primary tumor site.

| Cancer | # Selected Features | DP-SurVAE(Ours) | DS-SurvSHAP | DS-SurvLIME | DS-SHAP | DS-LIME |
|---|---|---|---|---|---|---|
| BLCA | 59.0 | **.661** | .567 | .587 | .635 | .581 |
|  | 5.2 | .03 | .03 | .03 | .03 | .03 |
| BRCA | 55.9 | .536 | .515 | **.566** | .528 | .456 |
|  | 3.4 | .03 | .04 | .03 | .02 | .04 |
| CESC | 56.8 | **.696** | .561 | .593 | .643 | .526 |
|  | 5.9 | .04 | .04 | .03 | .04 | .04 |
| COAD | 60.8 | **.613** | .554 | .607 | .552 | .570 |
|  | 4.6 | .04 | .04 | .04 | .04 | .05 |
| ESCA | 61.2 | **.652** | .587 | .635 | .620 | .645 |
|  | 5.9 | .03 | .04 | .04 | .02 | .02 |
| HNSC | 55.5 | **.579** | .536 | .542 | .554 | .544 |
|  | 5.9 | .03 | .03 | .02 | .03 | .03 |
| KIRC | 51.1 | **.713** | .601 | .591 | .653 | .536 |
|  | 5.7 | .01 | .03 | .02 | .02 | .02 |
| KIRP | 51.0 | .763 | .628 | .630 | **.792** | .747 |
|  | 4.9 | .04 | .06 | .07 | .04 | .03 |
| LGG | 58.3 | **.745** | .663 | .663 | .724 | .685 |
|  | 5.0 | .01 | .04 | .03 | .02 | .02 |
| LIHC | 61.0 | .538 | .533 | .536 | **.588** | .562 |
|  | 6.1 | .03 | .04 | .04 | .03 | .03 |
| LUAD | 50.9 | .593 | .548 | .565 | **.602** | .549 |
|  | 7.7 | .02 | .03 | .03 | .02 | .03 |
| LUSC | 58.7 | .544 | .537 | .491 | **.551** | .525 |
|  | 7.2 | .03 | .02 | .03 | .02 | .03 |
| MESO | 71.4 | .573 | .582 | .565 | .588 | **.602** |
|  | 4.2 | .05 | .06 | .05 | .04 | .03 |
| OV | 56.8 | .535 | **.539** | .504 | .531 | .495 |
|  | 10. | .03 | .02 | .02 | .02 | .02 |
| PAAD | 61.8 | .567 | .557 | .501 | **.600** | .547 |
|  | 7.4 | .05 | .05 | .05 | .05 | .04 |
| SARC | 55.7 | **.621** | .541 | .493 | .616 | .604 |
|  | 7.4 | .04 | .04 | .04 | .04 | .03 |
| STAD | 53.4 | **.588** | .575 | .526 | .545 | .550 |
|  | 5.9 | .02 | .03 | .03 | .03 | .04 |
| UCEC | 61.1 | **.629** | .523 | .552 | .547 | .526 |
|  | 4.0 | .03 | .06 | .04 | .03 | .06 |
| P-value |  | – | 4e-23 | 5e-23 | 3e-5 | 3e-19 |
| Rank |  | **1.61** | 3.89 | 3.72 | 2.28 | 3.5 |

Table 5: C-index comparison on miRNA cohorts with reduced columns (DP-SurVAE and DS variants only). Standard errors (2nd row), ranks and one-sided Wilcoxon p-values vs. DP-SurVAE (last rows).

2. Table 4 summarizes the number of instances and events ($\delta = 1$) for each cancer type in the miRNA dataset, including their TCGA identifier (ID) and primary tumor site.

3. Table 5 reports the C-index comparison on miRNA cohorts with reduced columns (DP-SurVAE and DS variants only). Standard errors (2nd row), ranks and one-sided Wilcoxon p-values vs. DP-SurVAE (last rows).

4. Table 6 reports the C-index comparison on mRNA cohorts; standard errors (2nd row), ranks and one-sided Wilcoxon p-values vs. DP-SurVAE (last rows); "DS" = DeepSurv post-trained.

5. Table 7 shows the time lapsed (in seconds) for the explanation methods in the mRNA cohorts.

| Cancer | # Selected Features | DP-SurVAE(Ours) | DS-SurvLIME | DS-SHAP | DS-LIME |
|---|---|---|---|---|---|
| BLCA | 911.2 | **.702** | .587 | .699 | .665 |
|  | 261 | .02 | .03 | .02 | .02 |
| BRCA | 914.4 | **.744** | .526 | .639 | .653 |
|  | 263 | .03 | .05 | .03 | .02 |
| CESC | 653.9 | .686 | .604 | **.721** | .701 |
|  | 181 | .04 | .04 | .03 | .04 |
| COAD | 685.9 | **.650** | .576 | .635 | .613 |
|  | 183 | .03 | .05 | .04 | .05 |
| ESCA | 485.4 | .519 | .525 | **.552** | .491 |
|  | 157 | .03 | .03 | .04 | .04 |
| GBM | 357.1 | .579 | .542 | .600 | **.605** |
|  | 113 | .04 | .04 | .03 | .04 |
| HNSC | 748.9 | **.649** | .538 | .633 | .631 |
|  | 214 | .02 | .03 | .02 | .02 |
| KIRC | 815.3 | .690 | .600 | **.710** | .687 |
|  | 269 | .02 | .03 | .01 | .02 |
| KIRP | 681.8 | **.735** | .706 | .727 | .716 |
|  | 161 | .08 | .08 | .06 | .06 |
| LGG | 901.3 | **.825** | .648 | .797 | .784 |
|  | 231 | .03 | .04 | .03 | .03 |
| LIHC | 865.0 | .598 | .551 | **.612** | .600 |
|  | 233 | .03 | .03 | .03 | .04 |
| LUAD | 711.2 | .630 | .542 | **.632** | .631 |
|  | 241 | .03 | .04 | .04 | .03 |
| LUSC | 994.1 | **.531** | .507 | .508 | .520 |
|  | 244 | .02 | .03 | .02 | .03 |
| MESO | 348.6 | .646 | .653 | **.687** | .656 |
|  | 74 | .04 | .03 | .03 | .03 |
| OV | 669.1 | **.612** | .512 | .592 | .584 |
|  | 187 | .01 | .02 | .02 | .02 |
| PAAD | 429.0 | **.699** | .591 | .689 | .656 |
|  | 107 | .02 | .04 | .03 | .03 |
| SARC | 643.2 | **.681** | .595 | .652 | .639 |
|  | 212 | .03 | .03 | .02 | .04 |
| STAD | 680.6 | .608 | .574 | .605 | **.620** |
|  | 281 | .03 | .04 | .04 | .04 |
| UCEC | 867.1 | .568 | .536 | .495 | **.570** |
|  | 240 | .02 | .05 | .04 | .05 |
| P-value |  | 8.5e-29 | 2.8e-36 | 1.3e-3 | 3.4e-8 |
| Rank |  | **1.84** | 3.79 | 1.95 | 2.42 |

Table 6: C-index comparison on mRNA cohorts; standard errors (2nd row), ranks and one-sided Wilcoxon p-values vs. DP-SurVAE (last rows); "DS" = DeepSurv post-trained.

6. Table 8 shows the time lapsed (in seconds) for the explanation methods in the miRNA cohorts.

7. Table 9 reveals the top 20 most significantly enriched (mean of corrected p-values) GO terms associated with the 3 000 highest weighted genes of DP-SurVAE.

8. Table 10 reveals the top 20 most significantly enriched (mean of corrected p-values) GO terms associated with the 3 000 highest weighted genes of SurvLIME.

9. Table 11 reveals the top 20 most significantly enriched (mean of corrected p-values) GO terms associated with the 3 000 highest weighted genes of SHAP.

10. Table 12 reveals the top 20 most significantly enriched (mean of corrected p-values) GO terms associated with the 3 000 highest weighted genes of LIME.

| Cancer | DP-SurVAE(Ours) | SurvLIME | SHAP | LIME |
|--------|------|----------|------|------|
| BLCA | 271.7 | 4006.5 | 24.3 | 13568.6 |
| BRCA | 592.2 | 4109.5 | 26.3 | 32683.6 |
| CESC | 189.8 | 3639.1 | 19.8 | 10016.9 |
| COAD | 235.6 | 2449.1 | 24.5 | 13285.4 |
| ESCA | 146.5 | 3177.2 | 10.2 | 5848.7 |
| GBM | 62.4 | 2671.5 | 7.7 | 4420.3 |
| HNSC | 284.8 | 3527.5 | 27.0 | 15349.4 |
| KIRC | 175.3 | 2403.1 | 22.7 | 14680.0 |
| KIRP | 115.1 | 3128.0 | 14.6 | 8316.6 |
| LGG | 361.9 | 3716.6 | 32.6 | 17341.8 |
| LIHC | 1610.0 | 2667.4 | 19.6 | 8771.3 |
| LUAD | 227.2 | 2841.5 | 24.5 | 15604.1 |
| LUSC | 357.3 | 3904.7 | 30.2 | 16054.1 |
| MESO | 58.4 | 2420.9 | 3.8 | 2395.8 |
| OV | 167.8 | 2820.5 | 19.3 | 10538.4 |
| PAAD | 180.3 | 2953.0 | 9.4 | 5076.2 |
| SARC | 152.9 | 3069.0 | 13.8 | 8197.7 |
| STAD | 261.2 | 3837.8 | 24.8 | 13466.7 |
| UCEC | 247.5 | 3745.2 | 24.5 | 14509.4 |

Table 7: Elapsed time (in seconds) for explanation methods across mRNA cohorts.

| Cancer | DP-SurVAE(Ours) | SurvSHAP | SurvLIME | SHAP | LIME |
|--------|------|----------|----------|------|------|
| BLCA | 28.4 | 1013.3 | 45.2 | 6.9 | 403.1 |
| BRCA | 54.6 | 1170.3 | 53.1 | 7.9 | 983.0 |
| CESC | 17.6 | 1070.1 | 43.0 | 4.8 | 289.8 |
| COAD | 195.5 | 1821.0 | 252.5 | 12.0 | 1124.3 |
| ESCA | 16.0 | 938.3 | 42.2 | 3.2 | 178.7 |
| HNSC | 32.8 | 1039.0 | 45.0 | 7.5 | 468.1 |
| KIRC | 102.7 | 1189.2 | 94.3 | 9.7 | 727.8 |
| KIRP | 16.7 | 1004.1 | 42.2 | 4.8 | 252.8 |
| LGG | 31.0 | 993.6 | 41.8 | 7.5 | 459.7 |
| LIHC | 24.2 | 921.3 | 40.5 | 5.6 | 312.8 |
| LUAD | 226.5 | 2233.9 | 454.4 | 18.1 | 1470.0 |
| LUSC | 26.6 | 1071.5 | 46.7 | 7.6 | 433.2 |
| MESO | 7.9 | 769.2 | 34.7 | 1.4 | 72.2 |
| OV | 28.9 | 998.4 | 42.8 | 7.5 | 417.7 |
| PAAD | 138.6 | 2373.0 | 462.2 | 7.8 | 630.0 |
| SARC | 16.0 | 940.9 | 40.5 | 4.1 | 219.5 |
| STAD | 24.5 | 954.2 | 44.6 | 7.0 | 397.4 |
| UCEC | 35.3 | 1106.5 | 51.8 | 8.4 | 543.6 |

Table 8: Elapsed time (in seconds) for explanation methods across miRNA cohorts.

| GO term | corr. p-value |
|---|---|
| signal transduction | 0.000006 |
| regulation of transcription by RNA polymerase II | 0.000089 |
| regulation of DNA-templated transcription | 0.000248 |
| negative regulation of transcription by RNA polymerase II | 0.001398 |
| cell differentiation | 0.001896 |
| immune system process | 0.002177 |
| positive regulation of transcription by RNA polymerase II | 0.002704 |
| G protein-coupled receptor signaling pathway | 0.003047 |
| positive regulation of DNA-templated transcription | 0.005269 |
| lipid metabolic process | 0.005456 |
| innate immune response | 0.006759 |
| cell adhesion | 0.009451 |
| apoptotic process | 0.010136 |
| monoatomic ion transport | 0.010327 |
| positive regulation of cell population proliferation | 0.010548 |
| transmembrane transport | 0.011755 |
| protein transport | 0.012208 |
| DNA damage response | 0.012284 |
| negative regulation of DNA-templated transcription | 0.014061 |
| intracellular signal transduction | 0.021577 |

Table 9: Top 20 most significantly enriched (mean of corrected p-values) GO terms associated with the 3 000 highest weighted genes of DP-SurVAE.

| GO term | corr. p-value |
|---|---|
| positive regulation of transcription by RNA polymerase II | 5.572385e-35 |
| positive regulation of DNA-templated transcription | 6.164510e-34 |
| signal transduction | 2.829912e-33 |
| intracellular signal transduction | 3.754660e-25 |
| cell differentiation | 5.119051e-25 |
| regulation of DNA-templated transcription | 1.771394e-24 |
| cell adhesion | 1.789299e-24 |
| regulation of transcription by RNA polymerase II | 2.825522e-24 |
| protein transport | 1.444897e-23 |
| apoptotic process | 1.000598e-22 |
| lipid metabolic process | 1.659174e-21 |
| monoatomic ion transport | 4.292173e-20 |
| negative regulation of transcription by RNA polymerase II | 4.755415e-20 |
| protein phosphorylation | 2.063416e-16 |
| protein ubiquitination | 2.171184e-16 |
| nervous system development | 3.341211e-16 |
| positive regulation of cell population proliferation | 1.852121e-15 |
| positive regulation of gene expression | 7.611382e-14 |
| cell migration | 7.778473e-14 |
| negative regulation of apoptotic process | 8.155840e-14 |

Table 10: Top 20 most significantly enriched (mean of corrected p-values) GO terms associated with the 3 000 highest weighted genes of SurvLIME.

| GO term | corr. p-value |
|---|---|
| immune response | 1.140518e-13 |
| inflammatory response | 9.357942e-12 |
| cell-cell signaling | 1.374636e-11 |
| nervous system development | 7.774137e-11 |
| antimicrobial humoral immune response mediated by antimicrobial peptide | 1.926833e-09 |
| signal transduction | 3.257645e-09 |
| cell adhesion | 4.555095e-09 |
| positive regulation of gene expression | 1.471969e-08 |
| cell differentiation | 5.476285e-08 |
| chemotaxis | 6.125405e-08 |
| extracellular matrix organization | 1.979039e-07 |
| positive regulation of cell population proliferation | 2.331090e-07 |
| killing of cells of another organism | 7.360719e-07 |
| proteolysis | 8.561255e-07 |
| cell surface receptor signaling pathway | 9.124258e-07 |
| chemokine-mediated signaling pathway | 1.900924e-06 |
| negative regulation of cell population proliferation | 8.903195e-06 |
| monoatomic ion transmembrane transport | 3.421388e-05 |
| transmembrane transport | 4.731696e-05 |
| positive regulation of phosphatidylinositol 3-kinase/protein kinase B signal transduction | 5.233693e-05 |

Table 11: Top 20 most significantly enriched (mean of corrected p-values) GO terms associated with the 3 000 highest weighted genes of SHAP.

| GO term | corr. p-value |
|---|---|
| antimicrobial humoral immune response mediated by antimicrobial peptide | 3.969633e-07 |
| signal transduction | 1.459593e-06 |
| cell adhesion | 2.140080e-06 |
| cell-cell signaling | 7.302211e-06 |
| cell differentiation | 1.025566e-05 |
| killing of cells of another organism | 3.235012e-05 |
| positive regulation of transcription by RNA polymerase II | 3.539859e-05 |
| immune response | 1.113486e-04 |
| negative regulation of transcription by RNA polymerase II | 2.375783e-04 |
| epithelial cell differentiation | 3.531834e-04 |
| inflammatory response | 3.603819e-04 |
| positive regulation of gene expression | 4.353938e-04 |
| regulation of DNA-templated transcription | 5.461153e-04 |
| monoatomic ion transport | 5.518000e-04 |
| nervous system development | 9.175706e-04 |
| positive regulation of cell population proliferation | 9.430675e-04 |
| anatomical structure morphogenesis | 9.772116e-04 |
| regulation of transcription by RNA polymerase II | 9.807193e-04 |
| defense response to bacterium | 9.959883e-04 |
| proteolysis | 1.105775e-03 |

Table 12: Top 20 most significantly enriched (mean of corrected p-values) GO terms associated with the 3 000 highest weighted genes of LIME.

| Cancer | DP-SurVAE (Ours) | SurvLIME | SHAP | LIME |
|--------|------|----------|------|------|
| BRCA | **0.63** | 0.53 | 0.55 | 0.56 |
|  | 0.01 | 0.01 | 0.01 | 0.01 |

Table 13: C-index comparison on mRNA BRCA METABRIC; standard errors (2nd row).

# E  BREAST CANCER (METABRIC)

We carefully re-examined the Breast Cancer (METABRIC, Nature 2012 & Nat Commun 2016)[11] dataset to ensure full anonymization in accordance with the strictest GDPR requirements. After obtaining the necessary legal approval, we evaluated our method on the METABRIC gene-expression data (20603 genes). We repeated all experiments five times, and the results (Table 13) below demonstrate that our method consistently outperforms the baseline approaches by a substantial margin.

## E.1  BIOLOGICAL ANALYSIS OF TOP-10 GO TERMS FOR EACH METHOD

For each method, the top-10 GO terms were selected based on their p-values:

- DP-SurVAE
    - regulation of transcription by RNA polymerase II,7.987514e-35
    - signal transduction,5.396556e-34
    - positive regulation of transcription by RNA polymerase II,9.676124e-26
    - regulation of DNA-templated transcription,2.672800e-24
    - cell differentiation,1.552473e-19
    - negative regulation of transcription by RNA polymerase II,1.552473e-19
    - G protein-coupled receptor signaling pathway,3.941524e-17
    - apoptotic process,1.235891e-16
    - nervous system development,2.293632e-16
    - DNA damage response,1.382339e-15
- SurvLIME
    - signal transduction,1.416493e-26
    - regulation of transcription by RNA polymerase II,4.806091e-19
    - regulation of DNA-templated transcription,2.696511e-18
    - positive regulation of transcription by RNA polymerase II,1.590828e-15
    - cell differentiation,3.208257e-14
    - protein ubiquitination,3.208257e-14
    - lipid metabolic process,6.890910e-14
    - G protein-coupled receptor signaling pathway,1.985122e-13
    - monoatomic ion transport,1.241796e-12
    - apoptotic process,1.008454e-11
- SHAP
    - signal transduction,4.825692e-66
    - G protein-coupled receptor signaling pathway,6.444372e-59
    - detection of chemical stimulus involved in sensory perception of smell,1.877696e-34
    - sensory perception of smell,4.307382e-33
    - regulation of transcription by RNA polymerase II,3.078287e-29

---

[11]https://www.cbioportal.org/study/summary?id=brca_metabric

- – cell differentiation,1.861322e-23
- – regulation of DNA-templated transcription,8.272321e-22
- – monoatomic ion transport,9.019970e-18
- – positive regulation of transcription by RNA polymerase II,1.485996e-17
- – lipid metabolic process,1.591441e-15
- LIME
  - – signal transduction,8.256411e-37
  - – regulation of transcription by RNA polymerase II,1.682700e-30
  - – negative regulation of transcription by RNA polymerase II,3.991687e-23
  - – regulation of DNA-templated transcription,8.770774e-23
  - – positive regulation of transcription by RNA polymerase II,4.495912e-20
  - – G protein-coupled receptor signaling pathway,3.198767e-18
  - – cell differentiation,4.669038e-18
  - – positive regulation of DNA-templated transcription,3.222355e-17
  - – cell adhesion,6.064152e-17
  - – apoptotic process,3.103161e-15

The most relevant biological processes for BRCA (breast cancer) are

1. DNA damage / homologous recombination repair defects (Royfman et al., 2021; den Brok et al., 2017; Liu & West, 2001): pathways like DNA damage response, DNA repair, cell cycle, transcription regulation, and apoptosis are highly characteristic.

2. Transcriptional dysregulation (den Brok et al., 2017; Scully, 2000): Breast cancer is strongly driven by transcriptional rewiring (e.g., ER/PR signaling, MYC amplification).

3. Cell differentiation changes (Scully, 2000; Ali et al., 2021): Loss of normal epithelial differentiation is a core phenotype.

4. GPCR or olfactory-receptor pathways (Chung et al., 2022; Masjedi et al., 2019): frequent false positives in GO analyses

| Method | Key Terms / Features | BRCA Relevance | Notes |
|--------|---------------------|----------------|-------|
| **DP-SurVAE** | DNA damage response, transcription regulation, cell differentiation, apoptosis, GPCR pathway | ++++ Extremely relevant | Strong biological alignment with BRCA; only method with DNA damage response |
| **SurvLIME** | Transcription regulation, metabolism, GPCR pathway | + Moderately relevant | Missing DNA damage response; weaker overall for BRCA |
| **SHAP** | Olfactory/sensory perception pathways, transcription regulation, cell differentiation | - Weak / artifact-prone | Dominated by olfactory/sensory terms (noise); not good for BRCA |
| **LIME** | Transcription regulation, cell differentiation, apoptosis | ++ Good | Missing DNA damage pathways; slightly weaker than DP-SurVAE |

Table 14: Comparison of key terms relevance for BRCA across interpretation methods.

Conclusion: DP-SurVAE is the most biologically plausible method for BRCA breast cancer analysis.

## F METHOD PARAMETERIZATION

Below, we detail the parameter settings used for DP-SurVAE (our method) and the baseline competitors:

- **DP-SurVAE (this work)**:

- Learning rate: $lr = 1 \times 10^{-4}$
- Latent space dimension: $l = 20$
- Desired sparsity level $\alpha$: 0.99 for miRNA, 0.999 for mRNA
- Mask regularization weights: $\beta = 10^3$, and $\lambda_1 = \lambda_2 = \lambda_3 = 1$

- **SurvLIME** (Kovalev et al., 2020):
  - mRNA: Number of samples $M = 5$, number of neighborhood samples $N = 10$
  - miRNA: Number of samples $M = 10$, number of neighborhood samples $N = 20$

- **SurvSHAP** (Krzyziński et al., 2023):
  - mRNA: Number of samples $M = 20$, number of background samples $N = 5$
  - miRNA: Number of samples $M = 20$, number of background samples $N = 5$

- **SHAP** (Lundberg & Lee, 2017):
  - We use `GradientExplainer`
  - mRNA: number of background samples $N = 100$
  - miRNA: the entire training set is used as the background

- **LIME** (Ribeiro et al., 2016):
  - We use `LimeTabularExplainer`
  - The entire training data is used as the neighborhood for both mRNA and miRNA

For each experiment and each fold, the fold number is used as a seed for the number generator.

## G METHOD PARAMETERIZATION

We conducted additional experiments, averaged over five runs across the 18 miRNA TCGA cohorts (Table 15), to assess the contributions of different components:

- Loss ablation: Removing the double-pass loss $L_{\text{dplpl}}$ reduces the C-index by 17% on average, confirming its substantial contribution.

- Masking ablation: Replacing the current masking scheme with random masking decreases the C-index by 11% on average.

- MMD vs. KL divergence: Substituting MMD with KL divergence results in a 6% drop in performance.

- $\mu$ vs. $z$: At test time, using $z$ from the parametrization trick instead of the mean leads to an 11% decrease in C-index. These results consistently highlight the importance of our design choices for robust survival prediction.

## H THE USE OF LARGE LANGUAGE MODELS (LLMS)

We have used ChatGPT as a general-purpose assisting tool: *(i)* for grammar checking of certain paragraphs, *(ii)* for discussing and better understanding related work, and *(iii)* for code formatting in a few Python modules.

| Cancer | No DPLPL | Z TRICK TEST | No MMD | Random Masking | DP-SurVAE(Ours) |
|--------|----------|--------------|--------|----------------|-----------------|
| BLCA | 0.483 | 0.553 | 0.640 | 0.576 | **0.661** |
|      | 0.04 | 0.05 | 0.03 | 0.05 | 0.03 |
| BRCA | 0.524 | 0.536 | 0.533 | **0.562** | 0.536 |
|      | 0.03 | 0.03 | 0.03 | 0.03 | 0.03 |
| CESC | 0.511 | 0.541 | 0.489 | 0.550 | **0.696** |
|      | 0.06 | 0.05 | 0.06 | 0.06 | 0.04 |
| COAD | 0.467 | 0.528 | 0.558 | 0.521 | **0.613** |
|      | 0.06 | 0.06 | 0.04 | 0.05 | 0.04 |
| ESCA | 0.516 | 0.598 | 0.526 | 0.527 | **0.652** |
|      | 0.07 | 0.05 | 0.05 | 0.06 | 0.03 |
| HNSC | 0.508 | 0.547 | **0.582** | 0.535 | 0.579 |
|      | 0.03 | 0.03 | 0.03 | 0.03 | 0.03 |
| KIRC | 0.503 | 0.564 | 0.704 | 0.605 | **0.713** |
|      | 0.05 | 0.05 | 0.01 | 0.05 | 0.01 |
| KIRP | 0.487 | 0.630 | 0.575 | 0.609 | **0.763** |
|      | 0.07 | 0.06 | 0.08 | 0.08 | 0.04 |
| LGG | 0.492 | 0.645 | 0.735 | 0.657 | **0.745** |
|     | 0.06 | 0.07 | 0.02 | 0.06 | 0.01 |
| LIHC | **0.543** | 0.517 | 0.514 | 0.523 | 0.538 |
|      | 0.03 | 0.04 | 0.03 | 0.04 | 0.03 |
| LUAD | 0.5227 | 0.551 | 0.592 | 0.529 | **0.593** |
|      | 0.03 | 0.04 | 0.02 | 0.05 | 0.02 |
| LUSC | 0.510 | 0.516 | 0.543 | 0.479 | **0.544** |
|      | 0.03 | 0.02 | 0.03 | 0.03 | 0.03 |
| MESO | 0.480 | 0.519 | 0.513 | 0.548 | **0.573** |
|      | 0.05 | 0.06 | 0.04 | 0.04 | 0.05 |
| OV | 0.515 | 0.519 | 0.525 | 0.492 | **0.535** |
|    | 0.02 | 0.02 | 0.02 | 0.02 | 0.03 |
| PAAD | 0.461 | 0.512 | 0.525 | 0.502 | **0.567** |
|      | 0.04 | 0.05 | 0.05 | 0.05 | 0.05 |
| SARC | 0.512 | 0.500 | 0.525 | 0.551 | **0.621** |
|      | 0.05 | 0.05 | 0.05 | 0.05 | 0.04 |
| STAD | 0.525 | 0.517 | 0.583 | 0.502 | **0.588** |
|      | 0.04 | 0.03 | 0.04 | 0.03 | 0.02 |
| UCEC | 0.483 | 0.535 | **0.685** | 0.545 | 0.629 |
|      | 0.04 | 0.05 | 0.03 | 0.05 | 0.03 |
| Wins | 1 | 0 | 2 | 1 | **14** |

Table 15: Ablation study: c-index comparison on miRNA cohorts; standard errors (2nd row).

