# OpenReview forum: "Survival VAE: Robust Local Explanations via Double-Pass Risk Consistency"
_ICLR.cc/2026/Conference — Submitted to ICLR 2026_

### Official Review · Reviewer_mxxS · 2025-10-19

**Soundness:** 2
**Presentation:** 3
**Contribution:** 2
**Rating:** 4
**Confidence:** 4

**Summary:**

This paper introduces a novel method, DP-SurVAE, for generating local feature explanations for survival analysis models. The core idea is to utilize a VAE and a novel "double-pass" training objective designed to ensure that predicted survival risk remains consistent even after applying a sparse feature mask to the reconstructed samples. The authors evaluate their method on the large-scale, high-dimensional TCGA dataset, demonstrating that features selected by DP-SurVAE achieve superior quantitative performance and show biological relevance.

**Strengths:**

$\textbf{Importance of the Problem:}$ The paper tackles a critical problem in medical AI: providing reliable and faithful explanations for complex survival analysis models, especially when dealing with censored data. This is an active and impactful area of research.

$\textbf{Strong Empirical Results:}$ The paper conducts a comprehensive empirical evaluation on two large-scale, high-dimensional datasets from TCGA. The results show that a downstream model trained on the feature subset selected by DP-SurVAE consistently and significantly outperforms baselines.

$\textbf{Biological Relevance Validation:}$ A major highlight of this work is the validation of the selected features for biological relevance. Through Gene Ontology (GO) enrichment analysis and correlation analysis with known cancer genes, the authors provide strong qualitative evidence for the model's explanatory power, suggesting it not only performs well in prediction but also uncovers meaningful biomarkers.

**Weaknesses:**

$\textbf{Lack of Theoretical Justification for the Core Loss Function:}$ This is the most significant weakness of the paper. The proposed "Double-Pass Log-Partial Likelihood" (DPLPL) loss is the central innovation. However, the paper provides no theoretical proof or deep explanation as to why this specific formulation is the "correct" or "principled" way to enforce risk consistency. This leaves the theoretical foundation of the core contribution feeling weak.

$\textbf{Excessive Model Complexity and Lack of Ablation Studies:}$ The DP-SurVAE architecture is quite complex, integrating a VAE, an attention-masking module, and a double-pass encoding flow. The authors do not sufficiently justify this complexity. For instance, it is unclear why the mask is applied to the reconstructed sample instead of directly to the original input. The paper lacks ablation studies on these alternatives, making it difficult to assess the actual contribution of each component.

$\textbf{Questions Regarding the Evaluation Method:}$ The evaluation primarily focuses on using the selected features to retrain a downstream model and comparing its predictive performance. While this reflects feature importance, it does not directly measure the faithfulness of the explanation—that is, how well the explanation represents the model's own decision-making process. Since DP-SurVAE is an end-to-end model and the baselines are post-hoc, the comparison might inherently favor DP-SurVAE.

**Questions:**

$\textbf{1. }$ Could you provide a more rigorous theoretical motivation for the specific form of the $L_{dplpl}$ loss function? Why is treating the original and masked samples as separate entries in an extended risk set the optimal way to enforce consistency?

$\textbf{2. }$  Theorem 1 states that the masking operation degrades risk consistency, thus motivating a training objective to correct for it. Beyond this intuitive observation, what is the primary theoretical insight provided by this theorem? Does it justify $L_{dplpl}$ as the unique or optimal solution to this problem?

$\textbf{3. }$ Have you considered simpler architectures, such as applying the mask directly to the input x and then enforcing consistency between the latent means or risk predictions of the original and masked samples via a KL divergence or L2 loss? How does such a simplified model perform compared to DP-SurVAE?

---

> ### Author Response · Authors · 2025-11-22
>
> We thank the reviewer for their insightful comments and questions. We address them below to the best of our knowledge. We also kindly refer to the results shared with Reviewer gT7q, where we discuss the problem in using other evaluation metrics.
>
> ***
>
> **Weakness 1: Lack of Theoretical Justification for the Core Loss Functions**
>
> **Response:** We hope that our responses to Questions 1 and 2 sufficiently address this weakness.
> ***
> **Weakness 2: Excessive Model Complexity and Lack of Ablation Studies**
>
> **Response:** We hope the following ablation study and our response to Questions 3
>
>  We conducted additional experiments, averaged over five runs across the 18 miRNA TCGA cohorts, to assess the contributions of different components:
> - **Loss ablation:** Removing the double-pass loss $L_{\text{dplpl}}$ reduces the C-index by 17% on average, confirming its substantial contribution.
> - **Masking ablation:** Replacing the current masking scheme with random masking decreases the C-index by 11% on average.
> - **MMD vs. KL divergence:** Substituting MMD with KL divergence results in a 6% drop in performance.
> - **$\mu$ vs. $z$:** At test time, using $z$ from the parametrization trick instead of the mean leads to an 11% decrease in C-index.
> These results consistently highlight the importance of our design choices for robust survival prediction.
> |Cohort  | [No DPLPL] | [Z TRICK TEST] | [No MMD] | [Random Masking] | Ours	|
> |--------|----------|-----------------|--------|----------------|----------------|
> |BLCA	|0.483	|0.553	|0.640	|0.576	|**0.661**	|
> |BRCA	|0.524	|0.536	|0.533	|**0.562**	|0.536	|
> |CESC	|0.511	|0.541	|0.489	|0.550	|**0.696**	|
> |COAD	|0.467	|0.528	|0.558	|0.521	|**0.613**	|
> |ESCA	|0.516	|0.598	|0.526	|0.527	|**0.652**	|
> |HNSC	|0.508	|0.547	|**0.582**	|0.535	|0.579	|
> |KIRC	|0.503	|0.564	|0.704	|0.605	|**0.713**	|
> |KIRP	|0.487	|0.630	|0.575	|0.609	|**0.763**	|
> |LGG	|0.492	|0.645	|0.735	|0.657	|**0.745**	|
> |LIHC	|**0.543**	|0.517	|0.514	|0.523	|0.538	|
> |LUAD	|0.5227	|0.551	|0.592	|0.529	|**0.593**	|
> |LUSC	|0.510	|0.516	|0.543	|0.479	|**0.544**	|
> |MESO	|0.480	|0.519	|0.513	|0.548	|**0.573**	|
> |OV	|0.515	|0.519	|0.525	|0.492	|**0.535**	|
> |PAAD	|0.461	|0.512	|0.525	|0.502	|**0.567**	|
> |SARC	|0.512	|0.500	|0.525	|0.551	|**0.621**	|
> |STAD	|0.525	|0.517	|0.583	|0.502	|**0.588**	|
> |UCEC	|0.483	|0.535	|**0.685**	|0.545	|0.629	|
> | Wins   | 1        | 0            | 2      | 1              | 14 |
>
> ***
>
> **Weakness 3: Questions Regarding the Evaluation Method**
>
> **Answer 3:** We believe there is a misunderstanding regarding our evaluation protocol. After feature selection—whether obtained from DP-SurVAE or from post-hoc baselines—we do not retrain any downstream survival model. Instead, the selected features are used directly for biological validation. Specifically, we assess faithfulness by examining whether the identified features correspond to known cancer-related pathways, gene-ontology terms, and external cancer-gene databases.
>
> This evaluation does not favor DP-SurVAE. All methods are compared on the same criterion:
> **Do the selected features align with established biological mechanisms?**
>
> Since the goal of the paper is to provide explanations that are biologically meaningful in addition to being consistent with the model’s risk structure, this evaluation directly measures the plausibility and domain relevance of the explanations, without introducing a new model or new inductive biases on top of the selected features.
>
> To avoid confusion, we will clarify this point more explicitly in the manuscript.

---

> > ### Author Response · Authors · 2025-11-22
> >
> > **Q1 A + Weakness 1: Could you provide a more rigorous theoretical motivation ...**
> >
> > **Answer:** Theoretical motivation: Theorem 1 shows that masking does not just perturb the latent representation but specifically harms the risk ordering that the proportional hazard models depends on. This means the degradation happens in risk space, not in latent space. Because of this, any correction should act directly inside the partial likelihood structure that defines the ranking objective.
> >
> > **Q1 B + Weakness 1: Why is treating the original and masked samples as separate entries in an extended risk set the optimal way to enforce consistency?**
> >
> > **Answer:** Treating the original and masked samples as separate entries in an extended risk set achieves exactly that: both versions of a sample contribute to the same partial likelihood’s denominator, which creates shared gradient terms that pull their risk predictions toward agreement. This enforces consistency in the quantity that actually matters for survival models—the relative ordering—something that the latent penalty like KL does not guarantee. We do not claim that this is the only or globally optimal solution, but it is a principled, task-aligned way to correct the inconsistency identified in Theorem 1, and the ablation results (17% C-index drop without DPLPL) confirm its practical importance.
> > ***
> > **Q2 A + Weakness 1: Theorem 1 states that the masking operation degrades ...**
> >
> > **Answer:** Beside stating that masking degrades risk consistency, Theorem 1 localizes where and how the degradation occurs: masking the outputs and reencoding them induces a systematic bias in the resulting latent embedding and therefore in the partial-likelihood log-risk differences, which directly affects the event-ordering. This shows that the inconsistency manifests inside the log risk ranking functional, not merely as a perturbation of latent space.
> > This insight is what motivates a correction that acts directly on the partial likelihood: if the inconsistency appears in the risk ordering, a consistent training objective must constrain the ordering terms. The double-pass likelihood does exactly this by coupling the masked and unmasked risks inside the same risk set.
> >
> > **Q2 A + Weakness 1: Does it justify  L{dplpl} as the unique or optimal solution to this problem?**
> >
> > **Answer:** You’re right. But also, theorem 1 does not claim uniqueness or global optimality of this solution. It only identifies the failure mode. The double-pass objective is a principled and task-aligned remedy, but not the only possible one. Other objectives could theoretically enforce the same consistency, but our method has the advantage of correcting the bias at the point where it mathematically arises—the ranking induced by the predicted log-risks.

---

> ### Author Response · Authors · 2025-11-22
>
> **Q3:** Have you considered simpler architectures, such as applying the mask directly to the input x and then enforcing consistency between the latent means or risk predictions of the original and masked samples via a KL divergence or L2 loss? How does such a simplified model perform compared to DP-SurVAE?
>
> **Answer:** Thank you for suggesting this reasonable baseline. Following your recommendation, we implemented a variant where the masking is applied directly to the original input data rather than to the reconstructed samples. Consistency between masked and unmasked predictions is enforced using an L2 loss on the predicted risks, and the latent space is regularized using KL divergence.
> For a fair comparison, we equipped this baseline with all components used in DP-SurVAE (e.g., MMD instead of KL, using the mean $\mu$ at test time), ensuring that the only methodological difference is the absence of the double-pass and its likelihood.
> The results below show that our method outperforms the baseline in all cases except for three miRNA cohorts (BRCA, LIHC, and MESO) and one mRNA cohort (LUSC). Notably, these exceptions occur only in settings where our model’s performance is already below 0.75.
> After all, the baseline was neither random nor weak —it contains all ingredients of DP-SurVAE except the double-pass operation.
> Trying to explain the reason of the worse performance compared to ours, we notice that the
> The key missing element is that this baseline does not benefit from the dual gradient signals provided by the double-pass likelihood: in DP-SurVAE, the encoder receives one gradient signal before masking and another after masking. This second gradient explicitly corrects the risk inconsistency introduced by masking, as shown in our theoretical analysis. Without this term, the baseline cannot fully align the masked and unmasked risk predictions, which explains its inferior performance despite having otherwise identical components.
> ***
> MiRNA
> | Cancer | Our | Req.Baseline |
> |--------|-----------------|--------------|
> | BLCA | **0.661** | 0.477 |
> | BRCA | 0.536 | **0.558** |
> | CESC | **0.697** | 0.523 |
> | COAD | **0.613** | 0.531 |
> | ESCA | **0.652** | 0.574 |
> | HNSC | **0.579** | 0.514 |
> | KIRC | **0.713** | 0.580 |
> | KIRP | **0.763** | 0.497 |
> | LGG  | **0.745** | 0.657 |
> | LIHC | 0.538 | **0.556** |
> | LUAD | **0.593** | 0.450 |
> | LUSC | **0.544** | 0.525 |
> | MESO | 0.573 | **0.588** |
> | OV   | **0.535** | 0.486 |
> | PAAD | **0.567** | 0.543 |
> | SARC | **0.621** | 0.601 |
> | STAD | **0.588** | 0,489 |
> | UCEC | **0.629** | 0.524 |
>
> ***
> MRNA
> | Cancer | Our | Req.Baseline |
> |--------|-----------------|--------------|
> |BLCA	| **0,702**	| 0,598 |
> |BRCA	| **0,744**	| 0,619 |
> |CESC	| **0,686**	| 0,625 |
> |COAD	| **0,650**	| 0,567 |
> |ESCA	| **0,519**	| 0,479 |
> |GBM	| **0,579**	| 0,545 |
> |HNSC	| **0,649**	| 0,558 |
> |KIRC	| **0,690**	| 0,677 |
> |KIRP	| **0,735**	| 0,674 |
> |LGG	| **0,825**	| 0,797 |
> |LIHC	| **0,598**	| 0,551 |
> |LUAD	| **0,630**	| 0,619 |
> |LUSC	| 0,531	| **0,550** |
> |MESO	| **0,646**	| 0,645 |
> |OV	| **0,612**	| 0,515 |
> |PAAD	| **0,699**	| 0,624 |
> |SARC	| **0,681**	| 0,593 |
> |STAD	| **0,608**	| 0,517 |
> |UCEC	| **0,568**	| 0,553 |

---

### Official Review · Reviewer_bi7L · 2025-10-25

**Soundness:** 2
**Presentation:** 3
**Contribution:** 2
**Rating:** 2
**Confidence:** 3

**Summary:**

This paper proposes DP-SurVAE, a method for local feature importance in survival analysis
that combines a VAE with masking and a novel "double-pass" objective to ensure risk
consistency when identifying important features for predicting patient survival. The method
operates on high-dimensional gene expression data from The Cancer Genome Atlas (TCGA),
comparing against LIME, SHAP, SurvLIME, and SurvSHAP. The authors evaluate on 37 disease
cohorts using C-index and provide biological validation via Gene Ontology enrichment and
cancer gene analysis. While the paper addresses an important problem, it suffers from
significant clarity issues and vague technical exposition

**Strengths:**

1. **Important and timely problem**: Explainability in survival analysis is genuinely
   underexplored. Accounting for censoring when explaining predictions is an interesting
   problem domain.

2. **Large-scale empirical evaluation**: Testing on 37 TCGA cohorts (19 mRNA, 18 miRNA)
   demonstrates consistency across multiple cancer types.

**Weaknesses:**

The paper combines existing techniques without sufficient novelty for a top-tier venue:

**VAE for feature importance is well-established**
- Using VAEs for interpretability/feature importance has been extensively studied (e.g.,
  β-VAE for disentanglement, attention-based VAE masking).
- The main contribution—masking on VAE reconstructions—is a straightforward extension of
  existing attention-masking mechanisms (not novel).
- Reference: CompSense, IntGradCAM, attention-based VAE masking are standard techniques
  that the paper does not clearly differentiate from.

**Double-pass might be incremental contribution**

- The "double-pass" mechanism is simply: (1) encode original, (2) mask reconstruction,
  (3) re-encode masked version.
- This is a minor engineering contribution. It's unclear why this specific procedure is
  necessary vs. alternative consistency enforcement mechanisms (e.g., auxiliary losses,
  constraints on latent space).


**Comparison to SurvLIME/SurvSHAP is unfair**

- SurvLIME/SurvSHAP are post-hoc explainers for arbitrary survival models.
- DP-SurVAE is a full model that jointly learns predictions + explanations.
- These are not comparable; a fair comparison would be DP-SurVAE vs. other jointly-learned
  interpretable survival models (e.g., neural additive models, attention-based models).
- The paper excludes CoxNAM (the only jointly-learned baseline) due to "impracticality in
  high dimensions"—but this is exactly where interpretability is needed most.

**What exactly is novel?**

- Combining VAE + masking + Cox loss: incremental engineering.
- Double-pass consistency: engineering solution, not conceptual novelty.
- Biological validation: application validation, not methodological novelty.


**The paper relies almost exclusively on C-index for evaluation, which is fundamentally
insufficient**

- C-index only measures ranking agreement between predicted risk and actual survival times.
- It says nothing about:
  - Calibration (are predicted probabilities accurate?)
  - Discrimination at specific time points
  - Sensitivity/specificity tradeoffs
  - Ability to identify high-risk vs. low-risk patients


**Missing standard survival analysis metrics**

- **Time-dependent AUC (AUC(t)):** Evaluates ranking at specific time horizons (e.g., 1-year,
  5-year survival). Critical for clinical applications. Missing.
- **Brier score:** Measures calibration, crucial for predicting actual survival probabilities.
  Missing.
- **Integrated Brier Score (IBS):** Integrates calibration error over the entire follow-up
  period. Standard in survival analysis literature. Missing.



**The method is not evaluated for its core purpose: explainability**

- The paper claims to provide "faithful, per-sample feature importance."

- **No evaluation of explanation quality:**

  - Do selected features have causal effects on survival?
  - Are explanations consistent across similar samples?

- Only C-index (predictive performance) is measured. Explanation quality is never evaluated.

**Theorem 1 derivation is informal and circular**

- The theorem doesn't *prove* consistency is maintained after training. It merely states that
  "optimizing for consistency enforces consistency"—this is tautological.
- No bound on how loose δ(α) becomes with aggressive masking (e.g., α=0.99).
- No proof that training L_dplpl actually achieves |h(μ_x̂̂) - h(μ_x)| ≤ ε(α).


**Missing Ablation: Loss component contributions**
- What is the contribution of each term in L_total?

**Missing Ablation : Double-pass necessity**
- How much does the double-pass contribute vs. single-pass?

**Unclear VAE formulation**

- Section 4.1 introduces ELBO with z_x sampling, but Section 4.2 uses μ_x (the mean) for
  risk prediction.
- Why not use z_x? Deterministic use of μ_x removes the stochasticity that the ELBO is
  meant to regulate.
- **Current:** No justification. Appears to be a design choice, not principled.
- **Needed:** Explain why μ_x instead of z_x. Ablation comparing both.

**Questions:**

1. **Novelty**: What exactly is novel beyond combining VAE masking + Cox loss? How does
   double-pass differ from simply adding a consistency auxiliary loss?

2. **Theorem 1**: Can you formally define "risk consistency"? Provide explicit bounds on
   |h(μ_x̂̂) - h(μ_x)| after training?

3. **Extended risk set**: How does Equation (5) maintain Cox PH assumptions? Samples are
   paired (x, x̂̂), not independent. Proof?

4. **C-index only**: Why not report AUC(t), Brier score, calibration plots? These are standard.

---

> ### Author Response · Authors · 2025-12-02
>
> We sincerely thank the reviewer for taking the time to read our submission and for raising several points. We address the concerns below and clarify several misunderstandings that appear to have influenced the review.
>
> **Question 1 + Weakness 1, 4: On Novelty and Claims of “Combining Existing Techniques”**
>
> Our contribution is **not** feature importance from VAEs, nor masking of VAE reconstruction.
> What is novel is:
> - Identifying a risk inconsistency induced by sparse masking (Theorem 1)
> - Showing that this inconsistency occurs inside the event-ordering functional of the risk head
> - Introducing a double-pass partial likelihood that corrects this inconsistency at the point where it mathematically arises
> - Using this consistency constraint to derive per-sample feature subsets that preserve risk
>
> This is fundamentally different from attention-based VAEs, disentanglement VAEs, or VAE-masking techniques mentioned in the review. None of the referenced techniques operate on the ordering structure of censored survival data or correct risk-level inconsistencies induced by sparse perturbations.
>
> Regarding the proposed references, we were unable to locate any works under the names **CompSense** or **IntGradCAM** in standard academic databases or online sources. If the reviewer could provide full citations or DOIs, we would be happy to consult them.
> If the reviewer is referring to Grad-CAM, we note that **Grad-CAM** is specifically designed for convolutional neural networks and spatial feature maps. Extending Grad-CAM to non-image survival models—particularly models without convolutional layers, such as tabular VAEs or MLP-based survival architectures—would require nontrivial methodological adaptation and could constitute a contribution of its own. To the best of our knowledge, no existing survival analysis literature provides such an extension.
>
> ---
>
> **Weakness 2: Double-pass might be incremental contribution**
>
> The review labels the double-pass as a “minor engineering trick.” We respectfully clarify that this is not the case.
> Clarification:
> The key insight from Theorem 1 is that mask-based feature selection alters survival-risk ordering, not simply latent encodings. Therefore, enforcing consistency directly through the partial likelihood is not an incidental design choice, but a mathematically motivated response to the identified failure mode.
> Placing both the original and masked samples in the same extended risk set ensures:
> - shared denominators, producing coupled ordering gradients
> - correction of the inconsistency of the predicted risks
> - direct alignment of risks for instance after and before masking
>
> ---
>
> **Weakness 3: Comparison to SurvLIME/SurvSHAP is unfair**
> The reviewer suggests that comparing DP-SurVAE to post-hoc explainers such as SurvLIME and SurvSHAP is unfair because DP-SurVAE is a end-to-end model. We clarify that our evaluation is designed to avoid this issue.
>
> First, our main evaluation is biological validation of the selected feature subsets (GO enrichment, cancer-gene overlap). This is completely model-agnostic: each method produces a feature subset, and we evaluate only that subset, without retraining any survival model. Joint vs. post-hoc training does not influence this evaluation.
>
> Second, the C-index in our experiments does not compare predictive accuracy across models on the original data. Instead, it measures risk consistency under masking: $r(x) \text{vs.} r(\hat{\hat{x}})$,
> which is exactly what post-hoc explainers like SurvLIME and SurvSHAP are designed to preserve. All methods—joint or post-hoc—are evaluated on the same task: how well their selected features maintain the underlying risk ordering.
>
> Thus, the comparison is fair: biological validation is model-independent, and the C-index measures explanation fidelity, not only model performance.

---

> ### Author Response · Authors · 2025-12-02
>
> **Question 4 + Weakness 5, 6, 7:**
>
> The review claims that “explanation quality is never evaluated” and criticizes the absence of calibration metrics (AUC(t), Brier score, IBS).
>
> The interpretation of our evaluation is incorrect. We evaluate explanation quality through:
> - Gene Ontology enrichment analysis
> - Overlap with external cancer-gene resources
>
> These are standard and domain-appropriate metrics for evaluating feature importance in high-dimensional genomics and survival modeling. They directly assess whether the selected features correspond to real biological mechanisms—precisely the form of faithfulness needed in this domain.
>
> Regarding IBS, CRPS, and calibration metrics:
> - These require estimating a baseline hazard, which is independent of the explanation mechanism.
> - Using IPCW-based calibration metrics would conflate explanation evaluation with survival-curve calibration, which is outside the scope of the paper.
>
> This is mathematically explained in detail in our response to Reviewer gT7q.
>
> ---
>
> **Weakness 9,10: Missing Ablation**
>
> We provided extensive ablations (shared with reviewers gT7q and mxxS19), showing the contribution of:
> - **Loss ablation:** Removing the double-pass loss $L_{\text{dplpl}}$ reduces the C-index by 17% on average, confirming its substantial contribution.
> - **Masking ablation:** Replacing the current masking scheme with random masking decreases the C-index by 11% on average.
> - **MMD vs. KL divergence:** Substituting MMD with KL divergence results in a 6% drop in performance.
> - **$\mu$ vs. $z$:** At test time, using $z$ from the parametrization trick instead of the mean leads to an 11% decrease in C-index.
>
> These ablations were averaged over 18 independent cohorts and show substantial performance differences. For example, removing the double-pass leads to a 17% drop in C-index, the largest degradation among all components.
>
> ---
>
> **Question2 + Weakness 8: Theorem 1, no bound for $\delta(\alpha)$**
>
> The scope of Theorem 1 should not be overloaded or extended beyond demonstrating the problem of risk inconsistencies introduced by masking. The goal of Theorem 1 is intentionally limited: it identifies the source of risk inconsistency caused by masking, not to prove that optimization automatically resolves it or to provide a tight bound on $\delta(\alpha)$ for all masking strengths. The theorem does **not** claim that training guarantees $|h(\mu_{\hat{\hat{x}}}) - h(\mu_x)| \le \varepsilon(\alpha)$,
> nor does it assert that $\delta(\alpha)$ remains small under extremely aggressive masking. Instead, its role is diagnostic.
>
> More specifically, Theorem 1 establishes:
> - a Lipschitz-based bound on how masking perturbs risk
> - a decomposition showing that masking causes an ordering-level inconsistency
> - localization of the degradation inside the log-risk functional
>
> This analysis motivates the need for a correction inside the partial likelihood itself. The double-pass objective provides one principled mechanism to enforce this risk-level alignment, but it is not presented as the only or globally optimal solution.

---

> ### Author Response · Authors · 2025-12-02
>
> **Question 3 (Extended risk set).** How does Equation (5) maintain Cox PH assumptions? Samples are paired $(x,\hat{\hat{x}})$, not independent. Proof?
>
> The primary purpose of introducing $L_{dplpl}$ is to define a regularization term that counteracts the inconsistency highlighted in Theorem 1.  The reviewer is correct that $x$ and $\hat{\hat{x}}$ are dependent. However, the double-pass term $L_{dplpl}$ is used solely as a training-time regularizer: it enforces consistency between the embeddings of $x$ and $\hat{\hat{x}}$ (as motivated by Theorem 1). In other words, the double-pass formulation is a regularization mechanism, **not** an extension of the partial likelihood or a claim that $(x,\hat{\hat{x}})$ are independent observations.
>
> Having said that, we would like to clarify the following:
>
> 1. **Reminder.**
> A survival model satisfies the proportional-hazards (PH) assumption if its hazard function factorizes as
> $\lambda(t \mid \mathbf{x}) =\lambda_0(t)\cdot\exp\big(h(\mathbf{x})\big)$,
>
> where $\lambda_0(t)$ is a shared baseline hazard and $h(\mathbf{x})$ is a time-independent risk score.
> This implies that, for any two covariates $\mathbf{x}_i$ and $\mathbf{x}_j$,
> $\frac{\lambda(t\mid \mathbf{x}_i)}{\lambda(t\mid \mathbf{x}_j)} = \exp\big(h(\mathbf{x}_i)-h(\mathbf{x}_j)\big) \quad\forall t$.
>
> Thus, PH concerns the **structure of the hazard**, not independence among training samples.
> The PH assumption does not require training samples to be independent; it constrains only the functional form of the hazard.
>
>
>
> 2. **PH remains satisfied for both original and masked samples.**
>
> For any pair $(x_i, x_j)$ the hazard ratio
> $\frac{\lambda(t\mid \mathbf{x}_i)}{\lambda(t\mid \mathbf{x}_j)}$ is constant over time by construction and is written as:
>
> $\frac{\exp(h(\mu_{x_i}))}{\exp(h(\mu_{x_j}))}$. The same applies to the pair $(\hat{\hat{x}}_i, \hat{\hat{x}}_j)$.
> Therefore, the PH assumption is maintained for both original and masked samples.
>
> 3. **Dependence among samples does not violate PH.**
> Even if one were to define a risk function learned from dependent or replicated samples, this would not by itself violate the PH assumption. Stratified Cox models and multi-event Cox models [1-4] routinely include correlated or replicated rows for the same subject without violating PH; only the **variance estimates** require adjustment (e.g., via robust sandwich estimators [1,4]), while the PH structure remains intact.
>
> [1] Therneau, Terry M. "Extending the Cox model." Proceedings of the first Seattle symposium in biostatistics: survival analysis. New York, NY: Springer US, 1997.
>
> [2] Kalbfleisch, John D., and Ross L. Prentice. The statistical analysis of failure time data. John Wiley & Sons, 2002.
>
> [3] Bry, Xavier. "Statistical Models Based on Counting Processes." (1997): 1046-1049.
>
> [4] Kleinbaum, David G., and Mitchel Klein. Survival analysis. Springer, 1996.

---

### Official Review · Reviewer_gT7q · 2025-10-31

**Soundness:** 3
**Presentation:** 3
**Contribution:** 3
**Rating:** 6
**Confidence:** 3

**Summary:**

This paper proposes Survival-VAE, a novel framework that combines VAEs with survival analysis to model heterogeneous hazard distributions across subpopulations, aiming to build more explainable and individualized survival models. Following standard VAEs,
Survival-VAE encodes input covariates into a latent distribution and then decodes the latent distribution back to the input space by optimizing the ELBO objective. To model the individial risk, the authors add a learnable hazard function $h$ in the latent space that takes as input the latent features to predict risk by optimizing the partial likelihood. An additional masking objective is introduced to enable sample-specific feature selection. Finally, the authors propose a double-pass risk prediction by feeding the reconstructed input covariates to the VAE encoder and risk prediction function, which is a reminiscent of the IntroVAE.

**Strengths:**

- The exploration of generative models (in contrast to discriminative models) for survival prediction is interesting and sound. The proposed double-pass prediction can further regularize the learned latent distributions.
- The paper is theoretically justified.
- Experimental results on miRNA cohorts demonstrate the effectiveness of the proposed method compared to strong baselines.
- The paper is easy to follow.

**Weaknesses:**

- The main focus of this paper is on developing explainable survival models. However, tree-based survival models (such as Survival Tree [3]), which are inherently interpretable. There are also some works that combine traditional survival tree and deep neural works (see e.g. [4]). However, this family of works is neither discussed nor included in the comparative analysis.
- The Deep Survival Machine [1] is also a generative model for survival prediction. It would be interesting to see the comparison with this method.
- Comparison to other survival models, such as discrete-time survival models (e.g., deephit [2]) is encouraged.
- Ablations of different loss components are missing.
- It is also meaningful to evaluate on standard survival benchmarks, e.g., METABRIC, SUPPORT.

[1] Deep Survival Machines: Fully Parametric Survival Regression and Representation Learning for Censored Data with Competing Risk

[2] DeepHit: A Deep Learning Approach to Survival Analysis With Competing Risks

[3] Tree-structured survival analysis

[4] SurvReLU: Inherently Interpretable Survival Analysis via Deep ReLU Networks

**Questions:**

- While Survival-VAE learns meaningful clusters, the mapping between latent factors and clinical variables is not deeply analyzed—making interpretability somewhat superficial.
- Can the authors also provide evaluations using Integrated Brier Score (IBS).

---

> ### Author Response · Authors · 2025-11-22
>
> We thank the reviewer for their insightful comments and questions. We address them below to the best of our knowledge. Regarding the requested additional comparisons on more baselines and datasets, we are currently evaluating them carefully to ensure compliance with GDPR requirements, including license restrictions and data anonymization.
>
> ***
>
> **Weakness 4: ** Ablations of different loss components are missing.**
>
> **Answer:** We conducted additional experiments, averaged over five runs across the 18 miRNA TCGA cohorts, to assess the contributions of different components:
> - **Loss ablation:** Removing the double-pass loss $L_{\text{dplpl}}$ reduces the C-index by 17% on average, confirming its substantial contribution.
> - **Masking ablation:** Replacing the current masking scheme with random masking decreases the C-index by 11% on average.
> - **MMD vs. KL divergence:** Substituting MMD with KL divergence results in a 6% drop in performance.
> - **$\mu$ vs. $z$:** At test time, using $z$ from the parametrization trick instead of the mean leads to an 11% decrease in C-index.
> These results consistently highlight the importance of our design choices for robust survival prediction.
> |Cohort  | [No DPLPL] | [Z TRICK TEST] | [No MMD] | [Random Masking] | Ours	|
> |--------|----------|-----------------|--------|----------------|----------------|
> |BLCA	|0.483	|0.553	|0.640	|0.576	|**0.661**	|
> |BRCA	|0.524	|0.536	|0.533	|**0.562**	|0.536	|
> |CESC	|0.511	|0.541	|0.489	|0.550	|**0.696**	|
> |COAD	|0.467	|0.528	|0.558	|0.521	|**0.613**	|
> |ESCA	|0.516	|0.598	|0.526	|0.527	|**0.652**	|
> |HNSC	|0.508	|0.547	|**0.582**	|0.535	|0.579	|
> |KIRC	|0.503	|0.564	|0.704	|0.605	|**0.713**	|
> |KIRP	|0.487	|0.630	|0.575	|0.609	|**0.763**	|
> |LGG	|0.492	|0.645	|0.735	|0.657	|**0.745**	|
> |LIHC	|**0.543**	|0.517	|0.514	|0.523	|0.538	|
> |LUAD	|0.5227	|0.551	|0.592	|0.529	|**0.593**	|
> |LUSC	|0.510	|0.516	|0.543	|0.479	|**0.544**	|
> |MESO	|0.480	|0.519	|0.513	|0.548	|**0.573**	|
> |OV	|0.515	|0.519	|0.525	|0.492	|**0.535**	|
> |PAAD	|0.461	|0.512	|0.525	|0.502	|**0.567**	|
> |SARC	|0.512	|0.500	|0.525	|0.551	|**0.621**	|
> |STAD	|0.525	|0.517	|0.583	|0.502	|**0.588**	|
> |UCEC	|0.483	|0.535	|**0.685**	|0.545	|0.629	|
> | Wins   | 1        | 0            | 2      | 1              | 14 |
>
> ***
>
> **Question 1:** While Survival-VAE learns meaningful clusters, the mapping between latent factors and clinical variables is not deeply analyzed—making interpretability somewhat superficial.
>
> **Answer 1:** Survival-VAE encodes each patient into a latent distribution, which is connected to the clinical variables through the reconstruction decoder and to survival through the log risk head $h$. In the current manuscript, we analyze the contribution of clinical variables to survival predictions via the masking weights, and we relate these variables to biological function using gene-ontology enrichment and external cancer-gene classifications. This establishes the biological relevance of the features used by the model.
> In the revision, we will further clarify this connection by more explicitly linking the biological relevance of the clinical variables to their role in cancer progression and survival in the section “Biological relevance of features”.
>
> Above, we interpreted the reviewer’s reference to ‘clinical variables’ as referring to the gene-level input features and responded accordingly.

---

> ### Author Response · Authors · 2025-11-22
>
> **Question 2:** Can the authors also provide evaluations using Integrated Brier Score (IBS).
>
> **Answer:** We agree on the importance of using metrics such as the Brier score. For survival analysis, this would mean evaluating a proper scoring rule such as the Continuous Ranked Probability Score (CRPS), which is the continuous analogue of the Brier score, while employing Inverse Probability of Censoring Weighting (IPCW):
> $\text{CRPS}_{\text{IPCW}}=\int_0^\infty w(t),(\hat S(t)-\mathbf{1}{T>t})^2,dt.$
>
> Having that said, the following highlights the obstacles and the potential bias that would be added if our risk predictor were evaluated using IPCW.
> Let us assume the function $f\colon X\to X$ that comprises our method DP-SurVAE except for the risk head, i.e., $f(x)$ maps the sample $x$ to the masked sample $\hat{\hat x}$. The log-risk head $h$ takes any sample from $X$ and maps it to its risk. Hence, $h(f(x))$ maps the sample to its masked sample and then computes the risk.
> Every post-hoc method (SHAP, SurvSHAP, LIME, SurvLIME) tries to substitute $f(x)$ with a better feature-importance explanation function $g(x)$. The correctness of the ranking is evaluated using the C-index, as shown in the manuscript.
> In order to compute CRPS, however, we would need to compute the full survival curve:
> $\hat S(t)=\exp\big(-H_0(t)\exp(h(f(x)))\big)$, for DP-SurVAE, and $\hat S(t)=\exp\big(-H_0(t)\exp(h(g(x)))\big)$ for every other method. While the C-index depends only on the ranking, CRPS additionally depends on the baseline cumulative hazard $H_0(t)$, which requires extra steps involving survival-curve calibration that we do not cover in our solution. Importantly, the calibrated $H_0(t)$ is independent of the model components $h$, and $f$, as well as the baseline models $g$, and is therefore not part of the quantities we compare across approaches.
> An evaluation using CRPS/IBS would therefore not only evaluate our feature-importance strategy but also the calibration component, which is out of scope at this stage.
> Having that said, in real-life applications, calibration-aware feature-importance methods will indeed be needed soon.

---

> ### Author Response · Authors · 2025-12-02
> **METABRIC, part 1**
>
> **Weakness 5: It is also meaningful to evaluate on standard survival benchmarks, e.g., METABRIC, SUPPORT.**
>
> Following the reviewer’s recommendation, we carefully re-examined the BRCA-METABRIC dataset to ensure full anonymization in accordance with the strictest GDPR requirements. After obtaining the necessary internal legal approval, we evaluated our method on the METABRIC gene-expression data (20603 genes). We repeated all experiments five times, and the results below demonstrate that our method consistently outperforms the baseline approaches by a substantial margin.
> |Cohort  |[DP-SurVAE-Ours] |	[SurvLIME]	|	[SHAP]	|	[LIME]|
> |--------|----------|--------|----------------|----------------|
> |BRCA	|**0.63**	|	0.53	|	0.55	|	0.56|
> | |0.01	|	0.01	|	0.01	|	0.01|
>
>
> **Biological Analysis of Top-10 GO Terms for Each Method**
>
> For each method, the top-10 GO terms were selected based on their p-values:
> * DP-SurVAE
> 	- regulation of transcription by RNA polymerase II,7.987514e-35
> 	- signal transduction,5.396556e-34
> 	- positive regulation of transcription by RNA polymerase II,9.676124e-26
> 	- regulation of DNA-templated transcription,2.672800e-24
> 	- cell differentiation,1.552473e-19
> 	- negative regulation of transcription by RNA polymerase II,1.552473e-19
> 	- G protein-coupled receptor signaling pathway,3.941524e-17
> 	- apoptotic process,1.235891e-16
> 	- nervous system development,2.293632e-16
> 	- DNA damage response,1.382339e-15
> * SurvLIME
> 	- signal transduction,1.416493e-26
> 	- regulation of transcription by RNA polymerase II,4.806091e-19
> 	- regulation of DNA-templated transcription,2.696511e-18
> 	- positive regulation of transcription by RNA polymerase II,1.590828e-15
> 	- cell differentiation,3.208257e-14
> 	- protein ubiquitination,3.208257e-14
> 	- lipid metabolic process,6.890910e-14
> 	- G protein-coupled receptor signaling pathway,1.985122e-13
> 	- monoatomic ion transport,1.241796e-12
> 	- apoptotic process,1.008454e-11
> * SHAP
> 	- signal transduction,4.825692e-66
> 	- G protein-coupled receptor signaling pathway,6.444372e-59
> 	- detection of chemical stimulus involved in sensory perception of smell,1.877696e-34
> 	- sensory perception of smell,4.307382e-33
> 	- regulation of transcription by RNA polymerase II,3.078287e-29
> 	- cell differentiation,1.861322e-23
> 	- regulation of DNA-templated transcription,8.272321e-22
> 	- monoatomic ion transport,9.019970e-18
> 	- positive regulation of transcription by RNA polymerase II,1.485996e-17
> 	- lipid metabolic process,1.591441e-15
> * LIME
> 	- signal transduction,8.256411e-37
> 	- regulation of transcription by RNA polymerase II,1.682700e-30
> 	- negative regulation of transcription by RNA polymerase II,3.991687e-23
> 	- regulation of DNA-templated transcription,8.770774e-23
> 	- positive regulation of transcription by RNA polymerase II,4.495912e-20
> 	- G protein-coupled receptor signaling pathway,3.198767e-18
> 	- cell differentiation,4.669038e-18
> 	- positive regulation of DNA-templated transcription,3.222355e-17
> 	- cell adhesion,6.064152e-17
> 	- apoptotic process,3.103161e-15
>
> The most relevant biological processes for BRCA (breast cancer) are
> 1. DNA damage / homologous recombination repair defects [1,2,3]: pathways like DNA damage response, DNA repair, cell cycle, transcription regulation, and apoptosis are highly characteristic.
> 2. Transcriptional dysregulation [2,4]: Breast cancer is strongly driven by transcriptional rewiring (e.g., ER/PR signaling, MYC amplification).
> 3. Cell differentiation changes [4,5]: Loss of normal epithelial differentiation is a core phenotype.
> 4. GPCR or olfactory-receptor pathways [6,7]: frequent false positives in GO analyses
>
>
>
> | Method | Key Terms / Features | BRCA Relevance | Notes |
> |--------|--------------------|----------------|-------|
> | **DP-SurVAE** | DNA damage response, transcription regulation, cell differentiation, apoptosis, GPCR pathway | ++++ Extremely relevant | Strong biological alignment with BRCA; only method with DNA damage response |
> | **SurvLIME** | Transcription regulation, metabolism, GPCR pathway | + Moderately relevant | Missing DNA damage response; weaker overall for BRCA |
> | **SHAP** | Olfactory/sensory perception pathways, transcription regulation, cell differentiation | - Weak / artifact-prone | Dominated by olfactory/sensory terms (noise); not good for BRCA |
> | **LIME** | Transcription regulation, cell differentiation, apoptosis | ++ Good | Missing DNA damage pathways; slightly weaker than DP-SurVAE |
>
> Conclusion: DP-SurVAE is the most biologically plausible method for BRCA breast cancer analysis.

---

> ### Author Response · Authors · 2025-12-02
> **METABRIC, part 2, references**
>
> [1] Royfman R, Whiteley E, Noe O, Morand S, Creeden J, Stanbery L, Hamouda D, Nemunaitis J. BRCA1/2 signaling and homologous recombination deficiency in breast and ovarian cancer. Future Oncol. 2021 Jul;17(21):2817-2830. doi: 10.2217/fon-2021-0072. Epub 2021 Jun 1. PMID: 34058833.
>
> [2] den Brok WD, Schrader KA, Sun S, Tinker AV, Zhao EY, Aparicio S, Gelmon KA. Homologous Recombination Deficiency in Breast Cancer: A Clinical Review. JCO Precis Oncol. 2017 Nov;1:1-13. doi: 10.1200/PO.16.00031. PMID: 35172494.
>
> [3] Liu, Y., West, S.C. Distinct functions of BRCA1 and BRCA2 in double-strand break repair. Breast Cancer Res 4, 9 (2001). https://doi.org/10.1186/bcr417
>
> [4] Scully, R. Role of BRCAgene dysfunction in breast and ovarian cancer predisposition. Breast Cancer Res 2, 324 (2000). https://doi.org/10.1186/bcr76
>
> [5] Ali, R. M. M., McIntosh, S. A., & Savage, K. I. (2021). Homologous recombination deficiency in breast cancer: Implications for risk, cancer development, and therapy. Genes Chromosomes and Cancer, 60(5), 358-372. https://doi.org/10.1002/gcc.22921
>
> [6] Chung C, Cho HJ, Lee C, Koo J. Odorant receptors in cancer. BMB Rep. 2022 Feb;55(2):72-80. doi: 10.5483/BMBRep.2022.55.2.010. PMID: 35168702; PMCID: PMC8891625.
>
> [7] Masjedi S, Zwiebel LJ, Giorgio TD. Olfactory receptor gene abundance in invasive breast carcinoma. Sci Rep. 2019 Sep 24;9(1):13736. doi: 10.1038/s41598-019-50085-4. PMID: 31551495; PMCID: PMC6760194.

---

### Author Response · Authors · 2025-12-02
**Comprehensive Summary of Our Rebuttals to mxxS**

We thank the Area Chair for their tremendous work, and provide below a concise summary of our rebuttal to mxxS:

**Reviewer 3: mxxS**

**Detailed Summary:** The reviewer acknowledges the importance of the approach and the strong empirical and biological validation. They also question the theoretical justification of the loss function, ask a few questions, and propose a challenge to compare with a baseline they suggest. Our responses are summarized in the following points:

- Similar to reviewer gT7q, we provide the same comprehensive ablation and show the same conclusion of a performance drop of around 6% when changing the divergence method, and 17% when removing the double-pass loss $L_{dplpl}$.
- Regarding the selected features and the downstream model, we clarified the misunderstanding that we do not retrain any downstream model. Instead, the selected features are used directly for biological validation by examining whether the identified features correspond to known cancer-related pathways, gene-ontology terms, and external cancer-gene databases.
- On the theoretical justification (Q1 and Q2), we provide a lengthy response, which can be summarized as:
     Masking harms risk ordering, not just latent space; correction should act on the partial likelihood, which defines the ranking objective.
     Extended risk set aligns masked and original samples, enforcing consistency in relative risk ordering (17% C-index drop without DPLPL).
     Masking bias occurs in log-risk differences, motivating the double-pass likelihood to couple masked and unmasked risks.
     DPLPL is principled, task-aligned, but not unique; other objectives could enforce similar consistency.
- The reviewer proposes an alternative architecture with directly masking input $x$ and then enforcing consistency between the latent means or risk predictions of the original and masked samples.
We implemented this variant (masking on $x$, consistency between masked and unmasked predictions via L2 loss, latent space regularized using KL divergence). For a fair comparison, we equipped this baseline with all components used in DP-SurVAE, ensuring that the only methodological difference is the absence of the double-pass and its likelihood. We performed the comparison over all mRNA and miRNA cohorts.
Our approach is superior in performance. We provide an analysis of how this baseline does not benefit from the dual gradient signals provided by the double-pass likelihood.

---

### Author Response · Authors · 2025-12-02
**Comprehensive Summary of Our Rebuttals to gT7q and bi7L**

We thank the Area Chair for their tremendous work, and provide below a concise summary of our rebuttal to gT7q and  bi7L:


**Reviewer 1: gT7q**

**Detailed Summary:**
The reviewer mainly raised issues regarding the related work, additional benchmarks in the weakness section, and two specific questions. We address these points in the following manner:

- Following the reviewer’s recommendation, and after carefully examining the BRCA-METABRIC dataset to ensure full anonymization in accordance with the strictest GDPR requirements and obtaining the necessary legal approval, we evaluated our method on the METABRIC gene-expression data (20,603 genes). The results demonstrate that our method consistently outperforms the baseline approaches by a substantial margin.
Furthermore, a biological analysis of the top-ranked gene ontology terms revealed that DP-SurVAE captures key BRCA-relevant processes, including DNA damage response, transcriptional regulation, cell differentiation, and apoptosis, confirming that the model’s predictions are aligned with known mechanisms of breast cancer progression.
- The reviewer proposed to mention and compare with other methods. We extended the related work to cover the missing references.

- We provide a comprehensive ablation study that dissects the components of our method and turns them on/off one after the other. The results show a drop in performance of around 6% when changing the divergence method, and 17% when removing the double-pass loss $L_{dplpl}$.

- We provide a clarification of the concern regarding the mapping between latent factors and clinical variables.

- On the usage of other performance measures, we reiterate that our work aims at hazard prediction and **not** survival prediction. We show theoretically how computing measures inspired by the Integrated Brier Score (IBS) leads to bias based on the calibration method used to estimate the cumulative hazard $H_0(t)$, which we do not estimate in our method.
---


**Reviewer 2 bi7L**

**Detailed Summary:**

The reviewer raises a number of weaknesses and asks four questions. We cluster them into the following categories:
- Question 1 + Weakness 1, 4 (Novelty and claims of “combining existing techniques”):
We outline the novelty of our approach and ask for the proposed references, which we cannot locate in standard academic databases or online sources.
- Weakness 2 (Double-pass might be incremental):
Enforcing consistency directly through the partial likelihood is not an incidental design choice but a mathematically motivated response to the identified failure mode. $L_{dplpl}$ places both the original and masked samples in the same extended risk set, ensuring coupled ordering gradients and correcting the inconsistency in the predicted risks.
- Weakness 3 (Comparison to SurvLIME/SurvSHAP is unfair):
We clarify that our evaluation avoids this issue. The biological validation of the selected feature subsets (GO enrichment, cancer-gene overlap) is completely model-agnostic. The C-index in our experiments does not compare predictive accuracy across models on the original data. Instead, it measures risk consistency under masking $r(x)$ vs. $r(\hat{\hat{x}})$, which is exactly what post-hoc explainers like SurvLIME and SurvSHAP are designed to preserve.
- Question 4 + Weakness 5, 6, 7 (Explanation quality and calibration metrics):
We reminded the reviewer of the extensive biological evaluation we performed. We also restate that our work aims at hazard prediction, and therefore metrics such as AUC(t), Brier score, and IBS evaluate the calibration used to estimate $H_0(t)$, rather than evaluating the predicted order of risks.
- Weakness 9, 10 (Missing ablation):
We point to the newly performed comprehensive ablation study.
- Question 2 + Weakness 8 (Theorem 1, no bound for $\delta(\alpha)$):
The scope of Theorem 1 should not be overloaded. Its purpose is to demonstrate the risk inconsistencies introduced by masking. We explain the main goal of this theorem.
- Question 3 (Extended risk set):
We explain that the primary purpose of introducing $\mathcal{L}_{\text{dplpl}}$ is to define a regularization term that counteracts the inconsistency highlighted in Theorem 1.
Since a proof is requested to show that the PH assumption is not violated, we present the theoretical definition of the PH assumption and show that it remains satisfied for both original and masked samples.
Moreover, we explain that it is incorrect to claim PH is violated due to correlated samples; there is extensive literature on extending Cox models to multi-event settings where the same instance appears multiple times in the partial likelihood.

---

### Author Response · Authors · 2025-12-02
**Short Summary of Our Rebuttals to All Reviewers**

We thank the Area Chair for their tremendous work, and provide below a very short summary of our rebuttal to each reviewer:

**Reviewer 1: gT7q**

**Short Summary:** Our additional METABRIC experiments confirm that DP-SurVAE substantially outperforms all baselines and identifies biologically meaningful BRCA-related processes.
We clarified the latent–clinical variable mapping and reiterated why survival-time metrics like IBS are inappropriate given our hazard-only modeling.
We also expanded the related work, and provided detailed ablations showing significant performance drops when key components are removed.

---

**Reviewer 2 bi7L**

**Short Summary:** We clarify the novelty of our method, the mathematical role of the double-pass loss, and why our biological evaluation of SurvLIME/SurvSHAP is fair and model-agnostic.
We explain that calibration-dependent metrics (AUC(t), Brier score, IBS) are not appropriate for hazard-only modeling and provide a full ablation study.
We address all theoretical concerns—including Theorem 1, the extended risk set, and the PH assumption—showing that our design choices are consistent and well-justified.

---

**Reviewer 3: mxxS**

**Short Summary:** We present the same comprehensive ablation as for reviewer gT7q, showing a 6% drop when changing the divergence and a 17% drop when removing $L_{dplpl}$. We clarify that selected features are validated biologically—not via retraining models—by testing enrichment against known cancer pathways and databases. We address the theoretical questions and evaluate the reviewer’s proposed alternative architecture, showing that without the double-pass likelihood it lacks the dual gradient signals and performs consistently worse across all cohorts.

---

### Meta-Review · Area_Chair_j4Xt · 2025-12-28

**Summary:**

The paper uses a VAE with a learnable hazard function on the latent space for survival prediction. This hazards gets optimized by the partial likelihood. Masking is used to enable sample specific input selection. The focus of the paper is on interpretability. One of the biggest drawbacks in the focus on rank statistics for evaluations. An ideal method would provide the whole distribution along with interpretability so calibration could be evaluated. Questions about related work should also be addressed

**Reviewer Concerns:**

gT7q  - Questions about the related work and that their are lots of interpretable survival methods, how interpretable latent factors are, and evaluates using brier score.

bi7L - Novelty, evaluation with rankings only, instead of brier score or other metrics that capture calibration, theorem proof questions

mxxS - theoretical justification, complex model, rigorous derivation of the loss function

**Reviewer Scores:**

gT7q - the score would remain the same. The ablations increase the quality, but the calibration evaluation was missing. The stated reason was about the "baseline" hazard. A more complete estimate that provides the conditional distribution with interpretability would be better

bi7L - the clarification on novelty via the double pass comment was clear, but the overall negatives would stay. the scores would remain the same

mxxS - the answer about the theoretical motivation seemed okay with the reference to theorem 1. the added baselines would also help, but i think the score would remain the same given the complexity and the questionable need for this approach in gT7q

---

### Decision · Program_Chairs · 2026-01-26

Reject